# Highlights on mantle deformation beneath the Western Alps with seismic anisotropy using CIFALPS2 data

Pondrelli Silvia[1], Salimbeni Simone[1], Confal Judith M.[1], Malusà Marco G.[2], Paul Anne[3], Guillot Stephane[3], Solarino Stefano[4], Eva Elena[4], Aubert Coralie[3], Zhao Liang[5]

[1] Istituto Nazionale di Geofisica e Vulcanologia, Sezione di Bologna, Bologna, Italy

[2] Department of Earth and Environmental Sciences, University of Milano-Bicocca, Piazza della Scienza 4, 20126 Milan, Italy

[3] Univ. Grenoble Alpes, Univ. Savoie Mont Blanc, CNRS, IRD, UGE, ISTerre, Grenoble, France

[4] Istituto Nazionale di Geofisica e Vulcanologia, ONT, Genova, Italy

[5] State Key Laboratory of Lithospheric Evolution, Institute of Geology and Geophysics, Chinese Academy of Sciences, Beijing, China

*Correspondence to*: Silvia Pondrelli (silvia.pondrelli@ingv.it)

**Abstract.** There are still open questions about the deep structure beneath the Western Alps. Seismic velocity tomographies show the European slab subducting beneath the Adria plate, but all these images did not clarify completely the possible presence of tears, slab windows, or detachments. Seismic anisotropy, considered as an indicator of mantle deformation and studied using data recorded by dense networks, allows a better understanding of mantle flows in terms of location and orientation at depth. Using the large amount of shear wave splitting and splitting intensity measurements available in the Western Alps, collected through the CIFALPS2 temporary seismic network, together with already available data, some new patterns can be highlighted and gaps left by previous studies can be filled. Instead of the typical seismic anisotropy pattern parallel to the entire arc of the Western Alps, this study supports the presence of a differential contribution along the belt, only partly related to the European slab steepening. A nearly NS anisotropy pattern beneath the external Western Alps, a direction that cuts the morphological features of the belt, is clearly found with the new CIFALPS2 measurements. It is however confirmed that the asthenospheric flow from Central France towards the Tyrrhenian Sea, is turning around the southern tip of the European slab.

## 1 Introduction

Seismic anisotropy has become a convincing study tool, mainly in the areas where recent and past geodynamic evolution have left their marks in the mantle deformation and its patterns (e.g. Long and Silver, 2009; Long, 2013 for a complete review). Several methods exist to measure seismic anisotropy. Depending on the used seismic phase, signal, and frequency, it is possible to measure seismic anisotropy at different depths and relate it to different parts of the Earth's structure, last but not least with seismic anisotropy tomography (e.g. Zhao et al., 2023). Local seismicity and surface waves are used to measure

crustal seismic anisotropy, usually attributed to fractures or/and the state of stress of crustal depths (i.e. Crampin and Peacock, 2008; Okaya et al., 2018). Using seismic signals that travel deeper, it is possible to sample the deformation at lithospheric-asthenospheric depths. For instance, Pn phases record the seismic anisotropy immediately below the Moho in the lithospheric mantle (i.e. Diaz et al., 2013). On the other hand, using core phases (SKS, SKKS, etc) that record information on the receiver branch of their path, we can sample seismic anisotropy that is thought to be mainly concentrated in the upper mantle.

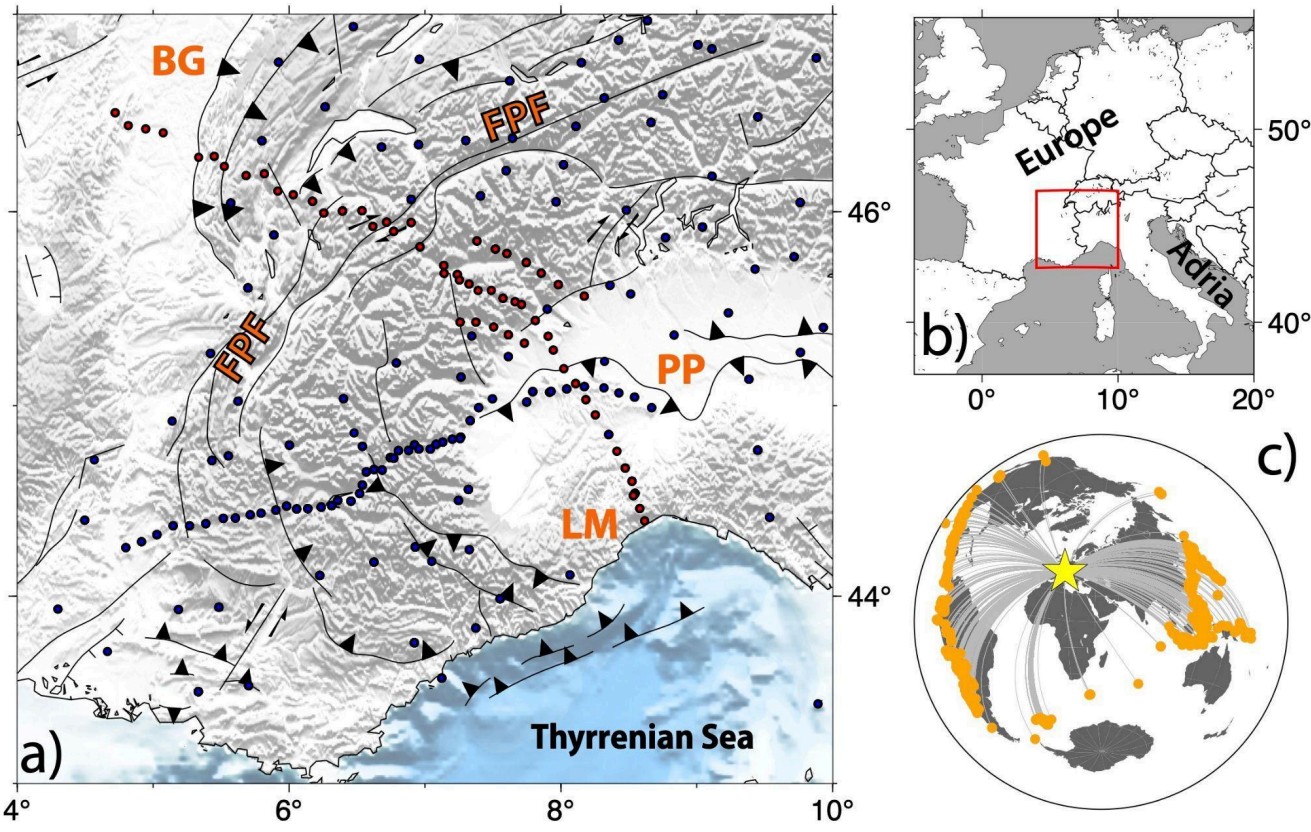

**Figure 1 - a) Map of the study region, focusing on the Western Alps. In red are indicated the CIFALPS2 stations, while in blue are permanent and previous temporary stations (i.e. CIFALPS and AlpArray). FPF = Frontal Pennine Fault, BG = Bresse Graben, PP = Po Plain, LM = Ligurian Mountains; b) the red square is the study area reported in a); c) map of all seismic events used in this study, with the star centered in the study region.**

The azimuth of the fast velocity direction and the delay time, the two parameters that commonly result from shear wave splitting analysis of core phases, are interpreted respectively as the direction assumed by olivine crystals, the principal mineral component of the upper mantle when mantle undergoes deformation, and the amount of anisotropy crossed by a seismic ray.

In the Alps these kinds of measurements improved immensely with recent temporary experiments such as AlpArray (Hetényi et al., 2018) or CIFALPS and CIFALPS2 in the western sector of the chain (Zhao et al., 2015; 2016: 2018), which complemented the permanent seismic networks operating in the region.

The European Alps originated in the late Cretaceous from the oblique subduction of the Alpine Tethys under the Adria microplate. The subduction evolved in a continental collision during the late Cenozoic (e.g., Handy et al., 2010, 2013 and references therein). In the Western Alps, the tectonic lineament that worked as the suture accommodating the shortening between the two plates is the Frontal Penninc Fault (FPF, Fig. 1). Even though the geological history of this belt is one of the best studied and well known in the world, the geodynamic evolution of the European slab, in terms of position and possible presence of slab break off, is still poorly understood.

All travel time tomographic studies identified at mantle depth the presence of seismic velocity heterogeneities interpreted as the European slab subducting beneath the Adria plate (e.g. Piromallo and Morelli, 2003; Lippitsch et al., 2003; Kissling et al., 2006; Giacomuzzi et al., 2011; Paffrath et al., 2021; Rappisi et al., 2022). However, the existence of possible slab detachments, windows or tears is debated, for instance beneath the Western Alps (e.g., Zhao et al., 2016). The first CIFALPS experiment (see Malusà et al., 2021 and references therein) clarified several points, starting from the first seismic evidence of subducted European continental lithosphere beneath the Adria lithosphere (Zhao et al., 2015), to a tomographic model with a continuous slab beneath this region (Zhao et al., 2016). In addition, recent seismic anisotropy analyses of the Western to the Central Alps shed additional light on potential discontinuities of the slabs, thanks to the possible mapping of mantle flows that would occur through them (Petrescu et al., 2020; Salimbeni et al., 2018).

The additional contribution of CIFALPS2, a temporary experiment deployed for 14 months from 2018 to 2019 (Zhao et al., 2018), on mantle seismic anisotropy mapping and interpretation, was expected to fill a gap in the northwestern part of the Alpine arc (red dots in Figure 1). Receiver function and ambient-noise tomography studies have underlined the north-south differences in the lithospheric structure along the belt strike (Paul et al., 2022). Therefore, there is a need for measuring additional seismic anisotropy in the mantle from CIFALPS2 data and to compare them to previous results.

In this study, we present the results of the analysis of data recorded by CIFALPS2, describing them in an integrated view with previous shear wave splitting  measurements (SWS) to identify new features in the mantle and draw hypotheses on their origin.

## 2 Data and Methods

Data used for the analysis are the recordings at CIFALPS2 stations (Figure 1; Zhao et al., 2018; doi: 10.15778/RESIF.XT2018) of teleseismic earthquakes with a magnitude M>6.0, that occurred between June 2018 and

December 2019 and are located at a distance interval from the network between 88° and 120°, typical to guarantee well isolated SKS phases in the waveforms. 80 to 150 events for each of the 56 temporary stations have been analyzed (Figure 1).

The entire SWS analysis has been conducted using the code SplitRacer (Reiss and Rümpker, 2017), based on the Silver and Chan (1991) method and thus on the minimization of the energy on the transverse component. Different filters have been applied according to the amount of noise at the various sites. For most of them, located in the Alps or Ligurian mountains, a bandpass filter between 7 and 20 s worked well, while for instance sites in the Po Plain needed different choices, i.e. 5-30 s. The signal to noise ratio (SNR) was also used to avoid noisy waveforms; initially, the threshold was 3, but where the amount of events to be analyzed was scarce we decreased it down to 1.5, again mainly for sites located in or close to the Po Plain. It is worth noticing that SWS analysis recovers fast-velocity directions, assuming a single layer of horizontal anisotropy. Moreover, the depth at which this anisotropy is located is difficult to define, but it is classically assumed that most measured anisotropy is in the upper mantle (Savage, 1999). Thus, it is common to visualize any lateral variation by plotting results at the piercing point of the incident ray at 150 km depth (Figure 2a).

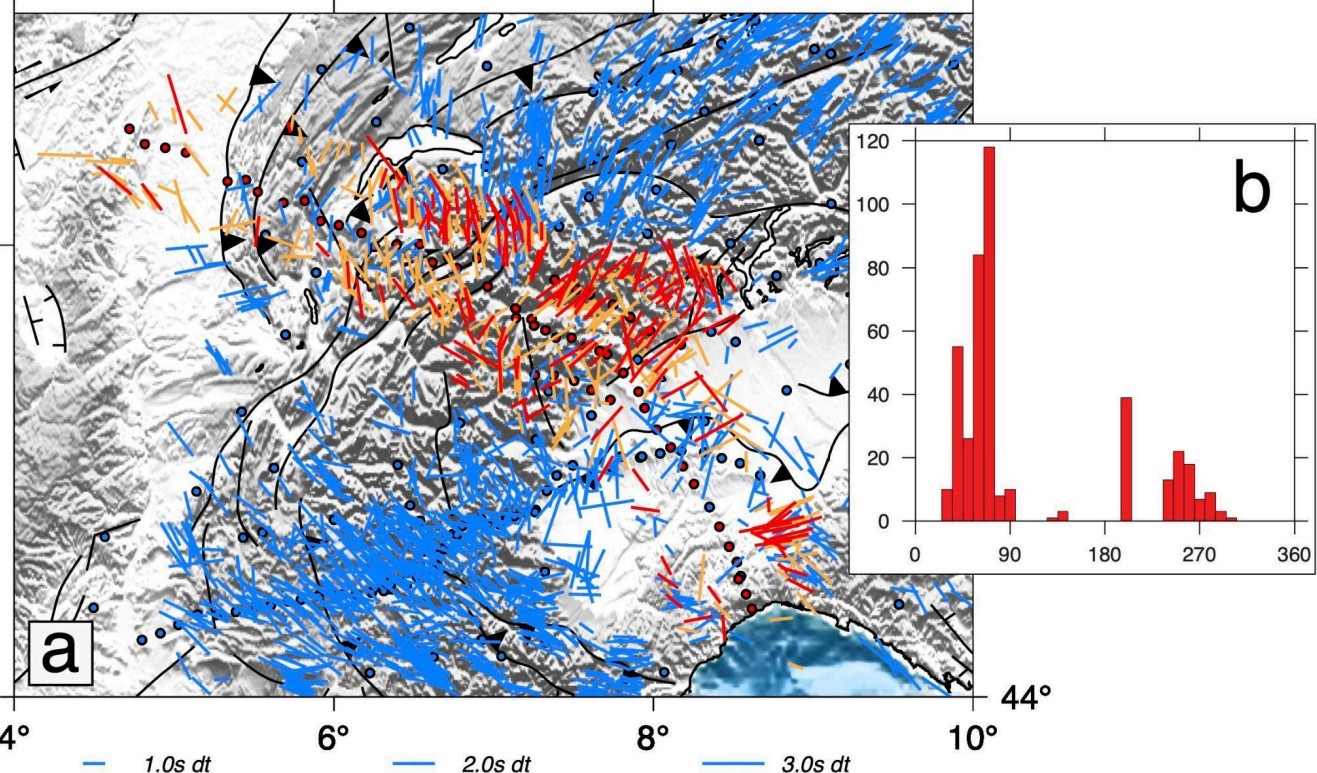

**Figure 2 - a) Map of single SWS measurements for the study region, plotted at the location of the piercing point of the ray at 150 km depth. In red and orange are good and fair measurements from CIFALPS2 stations (red circles) respectively. In the background, in light blue, previous SWS measurements and stations; b) histogram of back azimuths of events used in the analysis.**

To improve the resolution of the data to be discussed and interpreted, splitting intensity (SI) measurements have been performed on the same CIFALPS2 recordings used for SWS measures (Table S2 in Supplementary Material). Splitting intensity is measured by projecting the transverse component on the radial component derivative; it is related to the variations of the amplitude of the transverse component with the back azimuth (Chevrot, 2000; Monteiller and Chevrot, 2010). A routine based on Kong et al. (2015) and Confal et al. (2023) was used to calculate splitting intensity values from waveforms with a cut-off of 15 s before and 30 s after the supposed SKS arrival. A dominant period of 12 s is used for the Wiener filtering. To filter out low quality waveforms in this automatic process a cross-correlation coefficient of |0.7| and splitting intensities values and error threshold of |2.0| and 0.5 respectively were used (Baccheschi et al., under revision). With splitting measurements from at least four different 10° bins of back azimuths, the classical evaluation of the anisotropy parameters, i.e. the azimuth of the fast direction phi and the delay time dt, is obtained by fitting a sinusoid to the back azimuthal dependence of splitting intensity values (Chevrot, 2000). In particular, the sinusoid amplitude and phase give dt and phi respectively (see Figure S1 in Supplementary Material as an example).

## 3 Shear wave splitting results

From SWS analysis we obtained more than 400 pairs of splitting parameters (phi and dt) if we consider together good (170) and fair (241) results (Figure 2a, good in red, fair in orange; all results are available at https://osf.io/nqxk4, Pondrelli et al., 2023). The quality assignment is given following the SplitRacer criteria (Reiss and Rümpker, 2017), considering the visibility of the phase, the ellipticity of the initial particle motion and its linearity in the final stage, and the errors associated with phi and dt values. In Figure S2 of the Supplementary Material some measurement examples are shown. In addition, nearly 600 null measurements have been obtained (Figure S3 in Supplementary Material), where a null is considered when no split appears in the signal (i.e. no energy in the transversal component). This is due either to the absence of anisotropy or to the initial polarization being parallel to the fast or slow anisotropic direction. A high percentage of good and fair results were obtained for events with a NE back azimuth, so it should be taken into account that this direction is oversampled (Figure 2b).

This new dataset fills the region between the north-western external Alps and the Ligurian Sea. The anisotropy directions of no-nulls and nulls mostly agree with previous measurements (Figures 2 and 3, Figure S3 in Suppl. Material). Along the part of the transect crossing the Alps (transect AA' in Figure 3), NE-SW direction dominates in the internal part of the belt, between the FPF and the boundary of the Po Plain (see average values, red dots in Figure 3). In the western part of the AA' transect, measurements are more scattered, with a coexistence of NE-SW and NS to NNE-SSW directions. In the outer part of the Alpine belt and in the Bresse Graben, the prevailing directions are NS to NNE-SSW. Anisotropy in this region is weaker, but fast velocity directions remain constant toward the NW end of the transect, confirmed also by null measurements

(Fig.S3 in Supplementary Material). These two patterns, one NE-SW parallel to the Alps strike and the other nearly NS, and their location along the transect are well visible in Figure 3, mainly following average values (reddish dots).

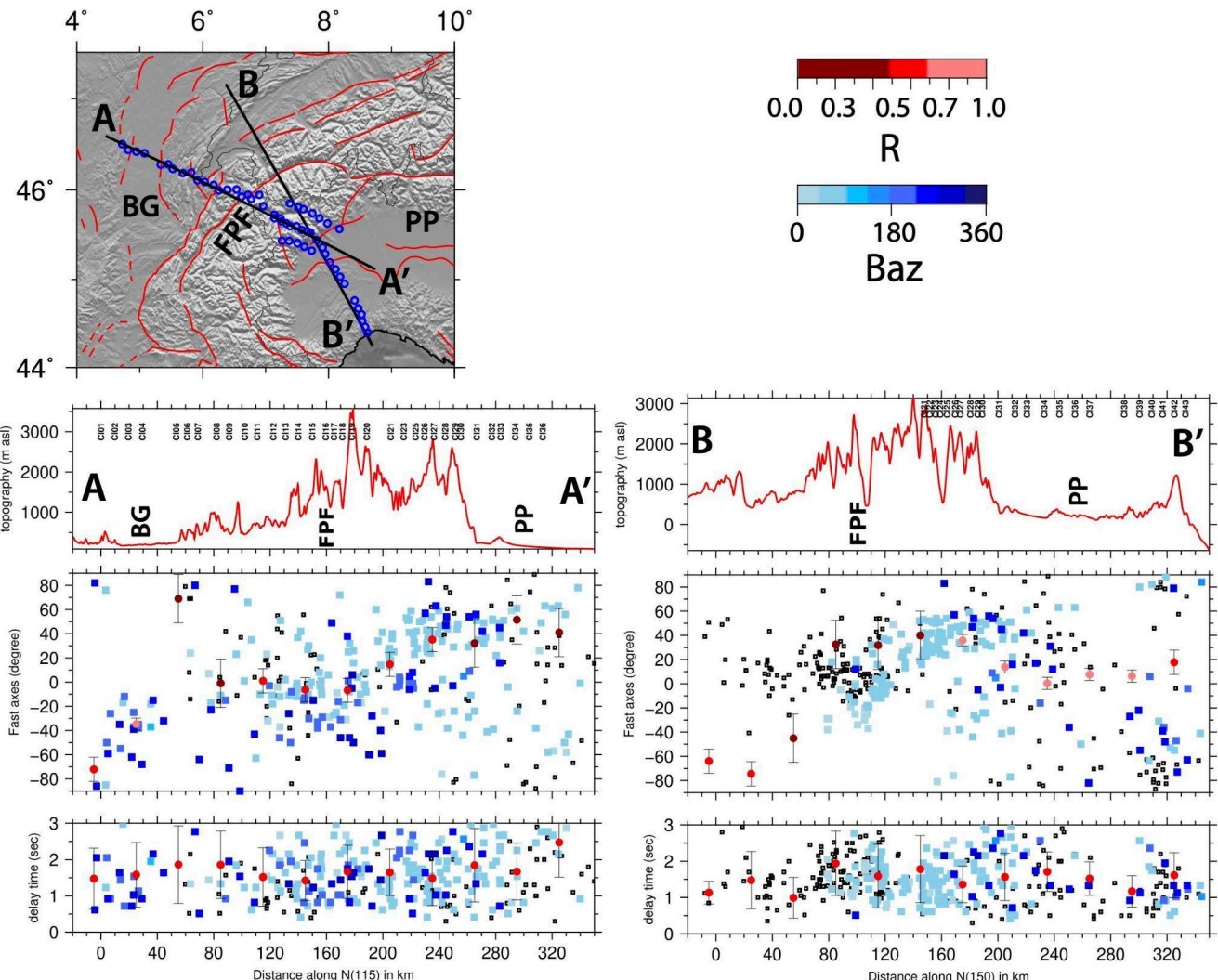

**Figure 3 - Distribution of splitting parameters along the two sections of the CIFALPS2 region. For each section topography (upper), fast axes direction (middle) and delay time (lower) distributions along a swath box of 30 km are displayed. In the fast axes and delay time panels, every single measurement is represented by a square; the results of CIFALPS2 are blueish and color coded in agreement with the back azimuth of the events analyzed, while the results of previous works are represented by smaller empty squares. In the fast axes panel, circles represent the average values calculated using a basic circular arithmetic mean inside the swath box with 30-km-step increment; they are coloured in agreement with the spreading distribution around the mean value (R=0 distribution completely scattered, R=1 distribution completely aligned with the mean direction). In the delay time panels, red dots are the average value and its error, calculated with arithmetic mean and standard deviation. FPF = Frontal Penninic Fault; BG = Bresse Graben; PP = Po Plain.**

In the Po Plain, measurements appear scattered, similar to previous studies (e.g. Salimbeni et al., 2018; Petrescu et al., 2020; Figures 2 and 3). In the transition between the belt and the plain, the typical NE-SW Alpine direction prevails. Even if measurements are scarce, this direction appears also in the center of the Po Plain, together with a few NW-SE directions located southeast, close to the Ligurian Alps. However section BB' in Figure 3 shows that anisotropy directions are diverse, resulting in a strong dispersion of average values computed along the section. In the Ligurian part of the transect, several directions are detected, ENE-SSW and NW-SE (Apenninic) on the eastern side with a few weaker (lower dt) NNE-SSW measurements in the western side.

In general, we do not find any particular pattern in the delay time measurements. Average values computed along the sections (Figure 3) are mostly constant, around 1.5 s with a large range in single measurement values.

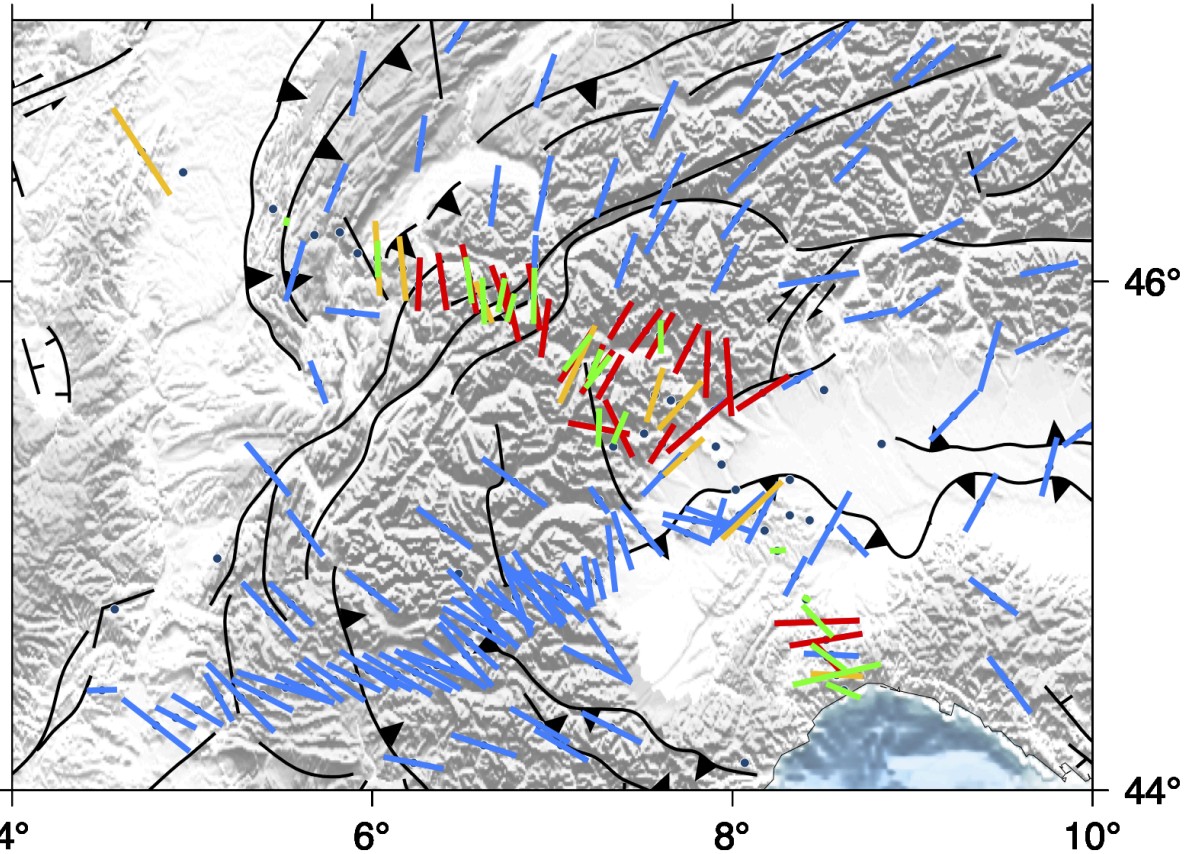

**Figure 4 - Maps of average SWS measurements (red, yellow and blue) and anisotropy parameters obtained using SI measurements (green). In red average values for CIFALPS2 stations obtained with more than 3 measurements, in yellow averages obtained with less than 3 values; in light blue, average measurements from previous works. Dots represent stations.**

In Figure 4, average SWS values for each station are mapped together with anisotropy values obtained by splitting intensity (SI) measurements (Table S1 in Supplementary Material). The first observation is that average SWS and SI data are very similar, mainly at sites where the single measurement results are more homogeneous, while differences are present where back azimuthal variations in fast direction are more evident, for instance in the southernmost part of the transect, in the Ligurian Mountains.

In general, it is clear that CIFALPS2 results are coherent with the average distribution of the anisotropy from previous measurements. The main Alpine pattern that follows the belt arc, here NE-SW, is represented in most of the averaged values for sites located in the transition between the Po Plain and the FPF. From the FPF to the NW endpoint of the transect, the main direction is close to NS, a direction that does not find an agreement with the orogen trend, still NE-SW.

In order to extract as much information as possible, we split the dataset into groups according to the main morphological features: a) External zone to the west of the FPF; b) Internal zone between the FPF and the western boundary of the Po plain, c) the Po Plain, and d) the Ligurian Alps. By plotting a rose diagram for all good and fair measurements obtained for the stations of each group, taking into account the influence of back azimuths, we get an overview of the main features and lateral changes of the anisotropy detected through these SWS measurements.

In Figure 5, the rose diagrams show evident differences along the CIFALPS2 transect, somewhere underlined by different patterns with respect to back azimuths. Only in group a), in the External zone, the N-S dominant direction remains, regardless of the amount of measurements or separation into opposite back azimuths (red and blue rose diagrams). For all other groups, there is a clear difference depending on the back azimuth. In group b) the entire dataset shows a main trend in a NE-SW direction, in agreement with the Alps strike in this region; the same direction is observed for events coming from NE (blue rose diagram). On the contrary, measurements obtained from events with a SW back azimuth are dominated by a nearly NS direction, quite similar to that shown by group a). The entire dataset for the Po Plain (c) and the subgroup with east back azimuths show a clear bimodal distribution, with both the Alpine NE-SW and the Apenninic NW-SE typical directions; west back azimuth events show instead N to NNW directions, very similar to those obtained in previous work at the first CIFALPS transect (e.g. Salimbeni et al., 2018) and again similar to group a) and SW back azimuth measurements of group b). In the Ligurian Alps, in the group d), opposite back azimuths show different results. It is worth noticing that in the group d) the ENE-WSW direction is dominant both in the all data plot and in the east back azimuth plot, while the west back azimuth plot shows a wide dispersion with a NW-SE direction prevailing.

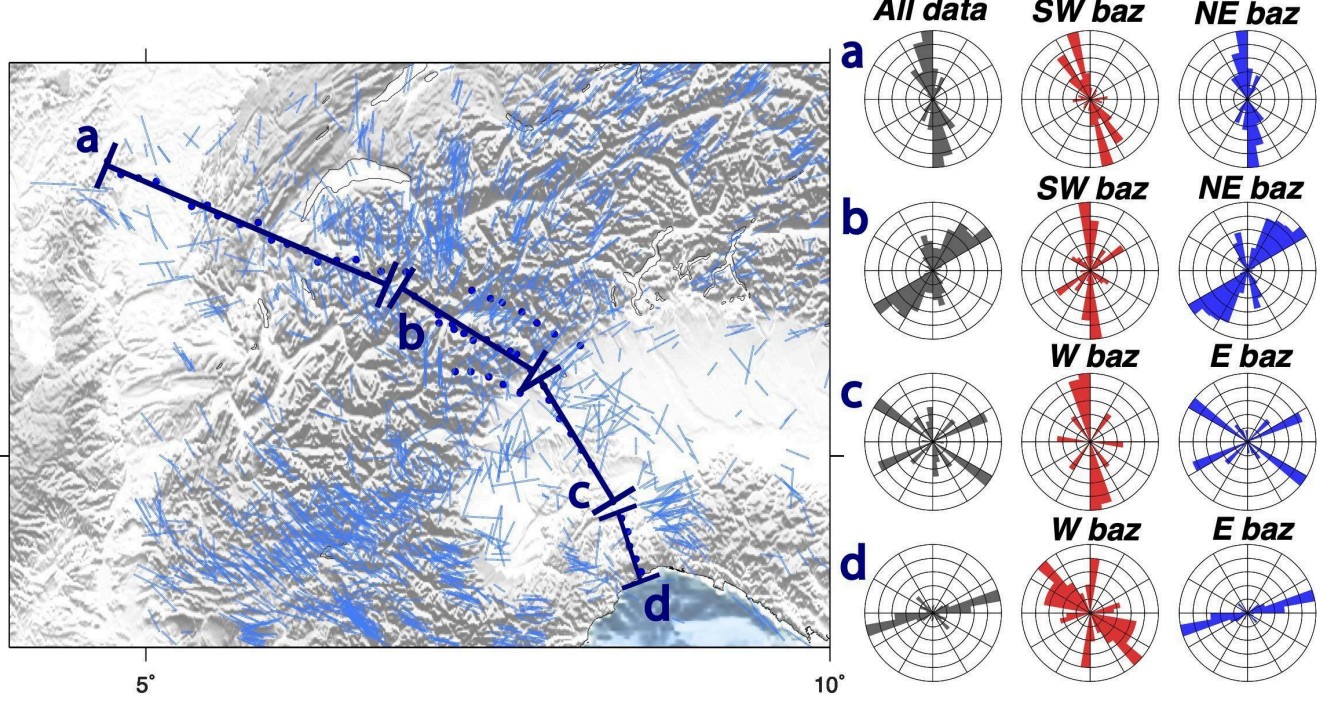

**Figure 5 - Normalised rose diagrams produced for groups of stations along the profile, a) External zone with respect to FPF; b) Internal zone with respect to FPF; c) Po Plain; d) the Ligurian Alps. Grey rose diagrams: all data; Red: events with SW (a,b) or W (c,d) back azimuths, blue: events with NE (a,b) or E (c,d) back azimuths.**

## 4 Discussion and hypotheses

CIFALPS2 SWS measurements clearly provide new information since they cover large areas where no seismic station is currently deployed or has been operating in the past. They allow us to fill data gaps and draw conclusions on the seismic anisotropy pattern and mantle deformation beneath the Western Alps.

A large-scale summary, prepared using the average values to avoid the scatter of single SWS measurements, is shown in Figure 6. ENE-WSW fast velocity directions, parallel to the strike of the Alps, are present from the Central to the Western Alps, in the transition between Po Plain and the belt, and terminate where the European slab and the belt are bending (double headed arrows in the inset sketch of Figure 6). At this point anisotropy directions do not strictly follow the chain strike, as they rather cut the main tectonic and morphological features. In the same zone (highlighted by the green circle A, Figure 6) converges a mantle flow that strikes from NE-SW to NS, in a coherent direction that however cuts the arcuate shape of the belt (yellow lines in the northern part of study region, Figure 6); this pattern is different from those described in previous anisotropy studies of the Western Alps (Barruol et al., 2004; Lucente et al., 2006; Barruol et al., 2011; Salimbeni et al., 2018)

where anisotropy has always been described as rotating with the belt direction. The discrepancy between the direction of tectonic lines and the nearly NS fast velocity is in agreement with a deep source of the anisotropy, in the mantle. Link and Rumpker (2023) in a recent analysis found that a similar nearly NS pattern would be mostly located in a shallower part of the mantle, and in part also present in a lower layer. They consider it as a shallow asthenospheric contribution.

On the contrary, the NW-SE asthenospheric flow identified from SE France toward the Ligurian Sea, which culminates in a flow around the southern tip of the European slab, is in agreement with previous studies. This flow apparently originates beneath Central France, in the Massif Central region (Barruol et al., 2004), and it seems that the NS mantle flow merges with it (left green circle, Figure 6) and then, flowing around the southern tip of the European slab, moves to the Tyrrhenian Sea.

The overlap with the teleseismic travel time tomographic image at 150 km depth by Zhao et al. (2016) indicates that all these fast velocity directions correspond to mantle deformation below the European slab. The mantle close to the slab has a NE-SW seismic anisotropy direction because it is deformed by the slab steepening. This feature is visible only in a narrow stripe, in the transition between Western and Central Alps, probably because in this place the slab steepening was favored by the plate motion direction (e.g. Adria anticlockwise rotation, Figure 6). Moving north, in the outer part of the belt, the anisotropy directions are those of an undeformed mantle, substantially parallel to the APM direction (orange arrows in Figure 6). Moving (south)westward, the deflection from NE-SW to NS may be related to the mantle being squeezed by the retreating European slab and moved toward the south where the retreat process was weakening or probably already ended. Indeed, in the Western Alps, the arcuate shape of the trench and slab, together with the (upper) Adria plate rotating in anticlockwise direction with a rotation pole more or less located in the western Po Plain (e.g. Serpelloni et al., 2016; Le Breton et al., 2017), may have been less favorable to the retreat of the slab. Such differences between the northern and southern Western Alps, in particular in the European Moho geometry, have been also identified by comparing CIFALPS and CIFALPS2 receiver function sections (Paul et al., 2022). The European Moho is strongly dipping down to ~75 km depth along CIFALPS section, while its dip is much weaker in CIFALPS2 section. Moreover, the absence of slab retreat in the Western Alps has been described also by Malusà et al. (2015), studying the mechanisms for exhumation of (U)HP terranes along the Cenozoic Adria-Europe plate boundary.

The mantle flow from beneath the Central Alps converges with the asthenospheric flow coming from SE France where Western Alps subduction, precisely a continental subduction (Malusà et al., 2021; Paul et al., 2022) runs out. In the region, the European slab is almost vertical (Zhao et al., 2016) and is no more affected by slab retreat. Substantially a differential behaviour of slab movement along the chain is at the base of the anisotropy pattern variation from Central to the entire Western Alps.

In the Po Plain, the image is patchy and complex, with very few good quality measurements and some really low values of dt. This does not necessarily mean that the mantle is isotropic, but that anisotropy may be multilayered or fast velocity directions are vertical.

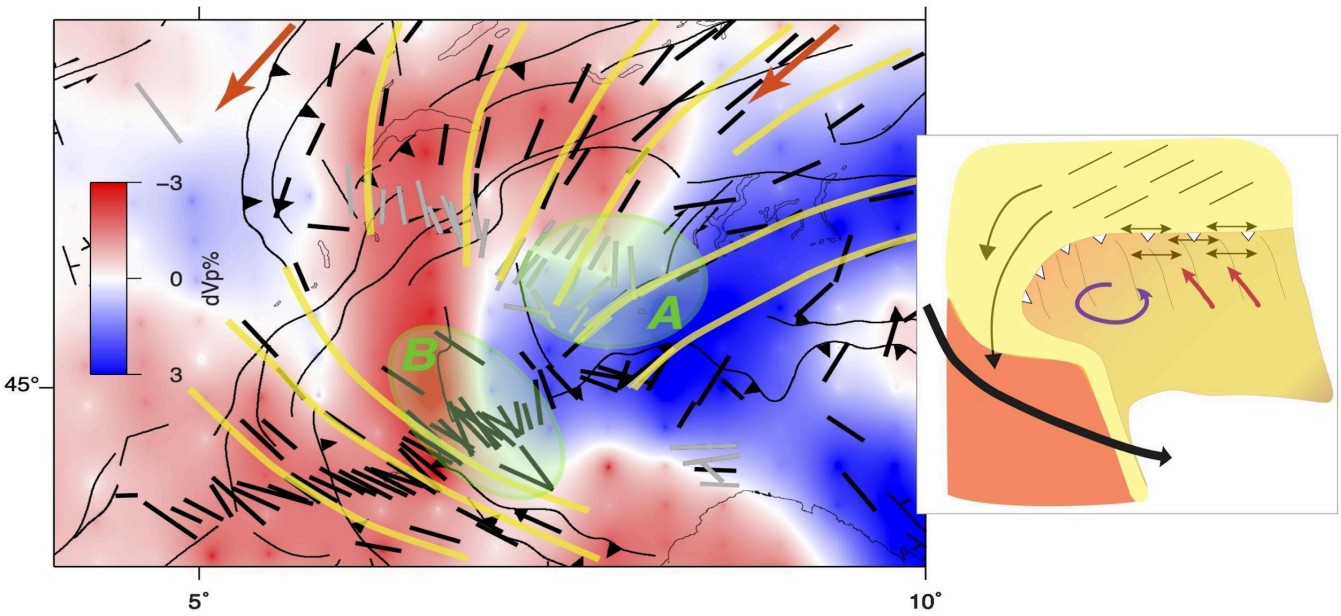

**Figure 6 - Left: Zhao et al. (2016) tomography (dVp in %) - layer at 150 km depth - overlapped with average SWS measurements: results from CIFALPS2 data in light grey and from previous studies in black. Orange arrows on top represent the absolute European plate motion from GSRM v2.1 model (calculated from Plate Motion Calculator | Software | GAGE). Yellow traces simulate average mantle deformation directions and green areas A and B highlight points where directions converge. Right: a sketch of this part of alpine subduction, where, in light yellow is the European plate, the double-headed arrows represent the only anisotropy of the study region related to European slab retreat (whose direction is represented by the red arrows), thin black lines are the anisotropy parallel to the APM, the thick black arrow represents the SE France asthenospheric mantle flow, the purple circular arrow represents the Adria plate rotation.**

It is worth noting that beneath the Po basin, the European slab is nearly vertical (e.g., Zhao et al. 2016; Paffrath et al., 2021) and the space for the mantle above is really narrow. With such a complex mantle structure, it is not surprising that a unique and significant pattern of anisotropy cannot be identified. In the Ligurian Alps, our measurements show different orientations east and west of the transect with, in general, a prevailing EW direction (Figure 5d), but a minor NNE-SSW set of measurements from the western back azimuths. These results are certainly intriguing but not sufficient to support any new hypothesis.

**5 Conclusion**

New data collected by the CIFALPS2 project clearly fill the gap that has forced the interpolation between more sparse information in the past. The pattern of seismic anisotropy shown here, is not entirely parallel to the belt, as it is in the Central Alps and in the transition to the Western Alps, up to the point this portion of belt is arcuate. Indeed, in the central part of the study region, a nearly NS pattern coming from central Europe cuts all principal morphologic features of the belt. It converges

with the part of the mantle deformed by the retreating slab in the point where the retreat is less favored or ended. The arcuate shape of the belt and of the slab, added to the Adria plate rotation with respect to Eurasia around a pole here particularly close to the boundary, reduce the effectiveness of a slab retreat. The NS mantle flow is then interpreted as the European mantle moving south to merge with the large asthenospheric flow that from beneath Central France moves toward the Tyrrhenian Sea turning around the southern tip of the European slab.

## Data availability

All shear wave splitting measurements from this work have been included and are available in the Italian shear wave splitting collection https://osf.io/nqxk4 (Pondrelli et al., 2023).

## Author contribution

PS, SS and CJM made measurements and analyses. PS prepared the manuscript with the contribution of all co-authors. All authors designed the CIFALPS2 experiment and carried it out.

## Competing interests

The authors declare that they have no conflict of interest.

## Acknowledgements

This research is funded by the National Natural Science Foundation of China and by NEWTON (NEw Window inTO Earth's iNterior), ERC StG funded project (grant ID:758199).

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

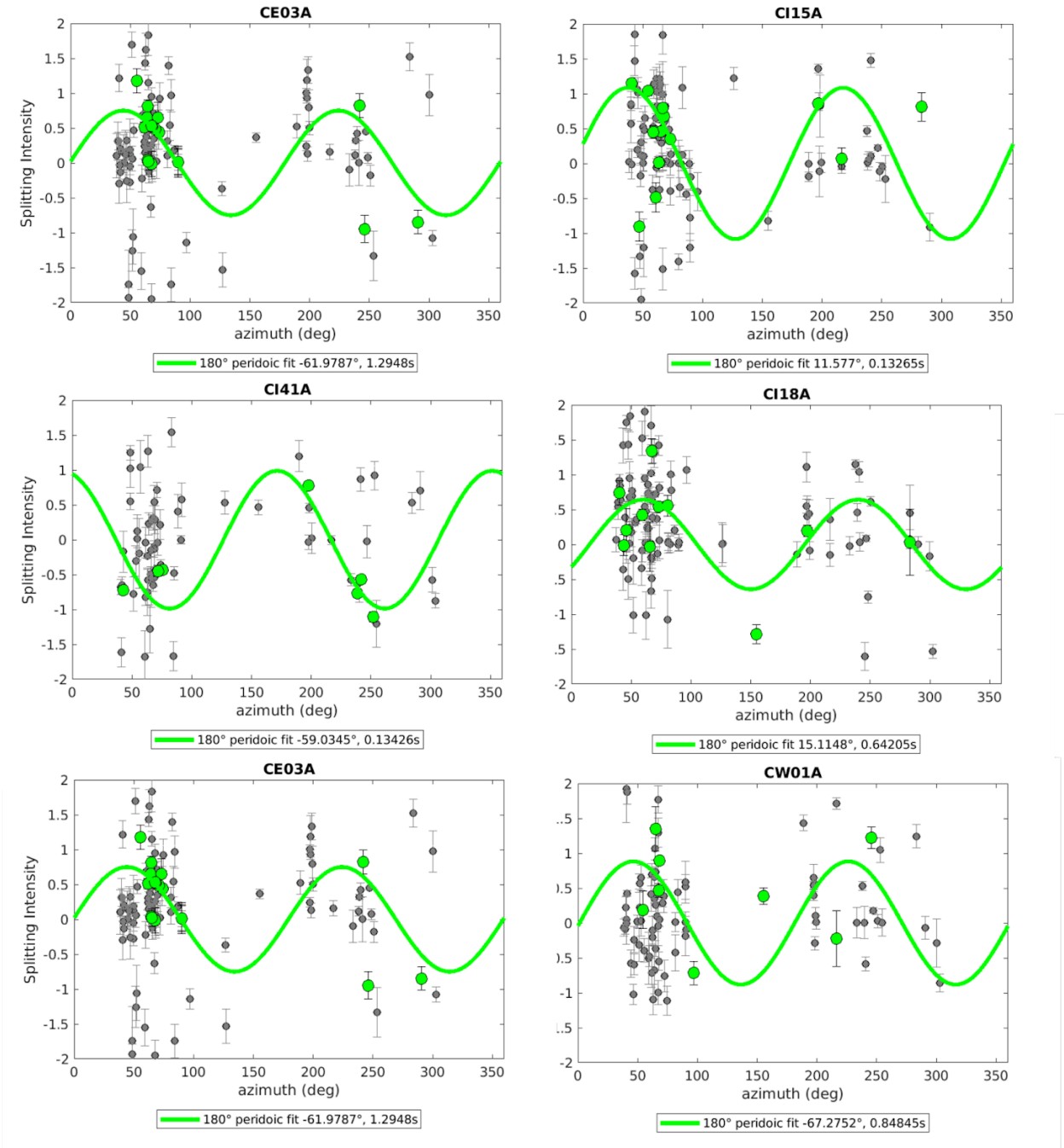

Figure S1 - Splitting intensity measurements for a few example stations. Gray dots: all measurements; green circles: only measurements that fit the criteria and the sinusoidal curve (fast polarization direction and time delay written below).

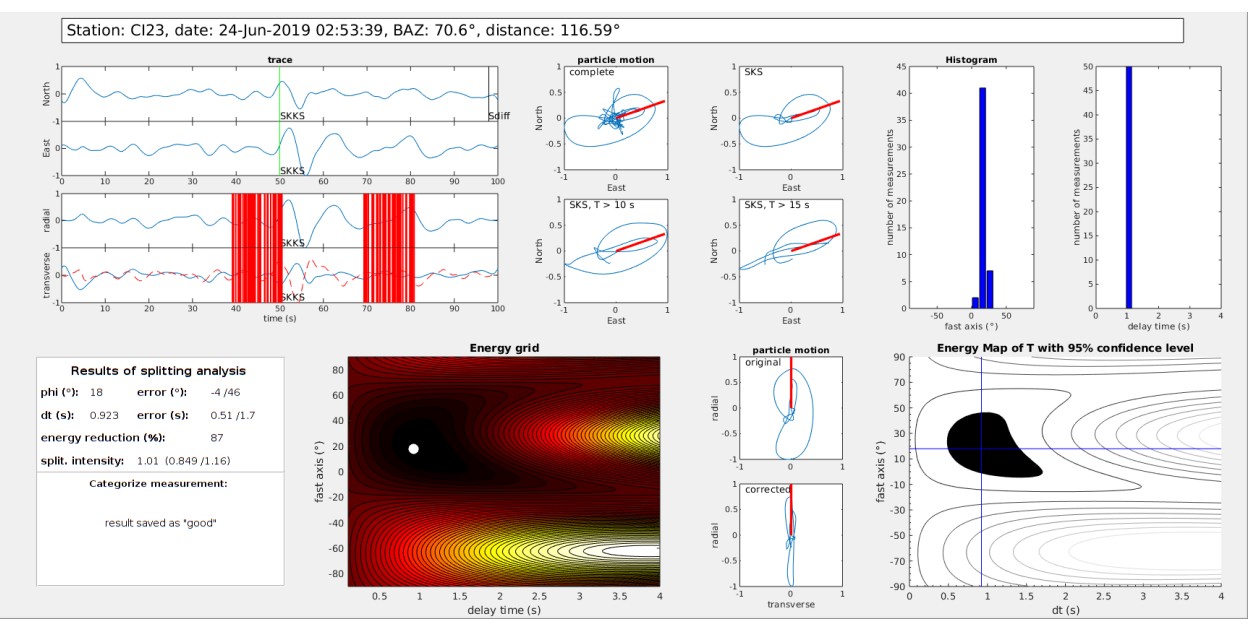

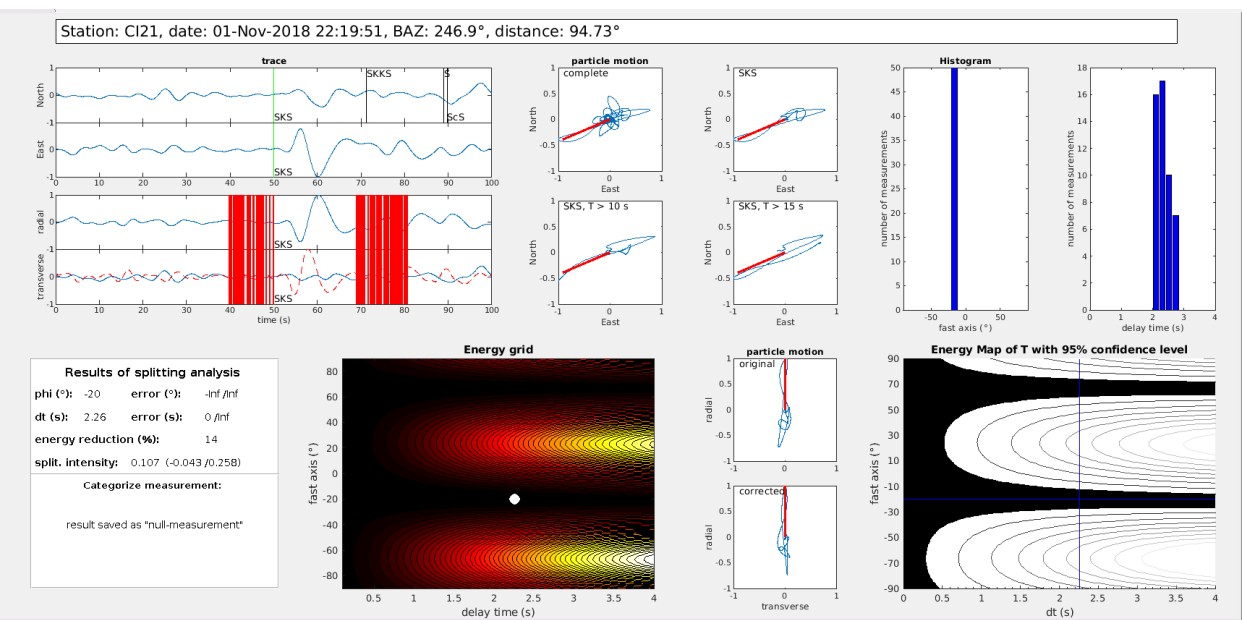

Figure S2 - Examples of measurements. Upper panel: a good shear wave splitting measurement at station CI23. Lower panel: a null measurement at station CI21.

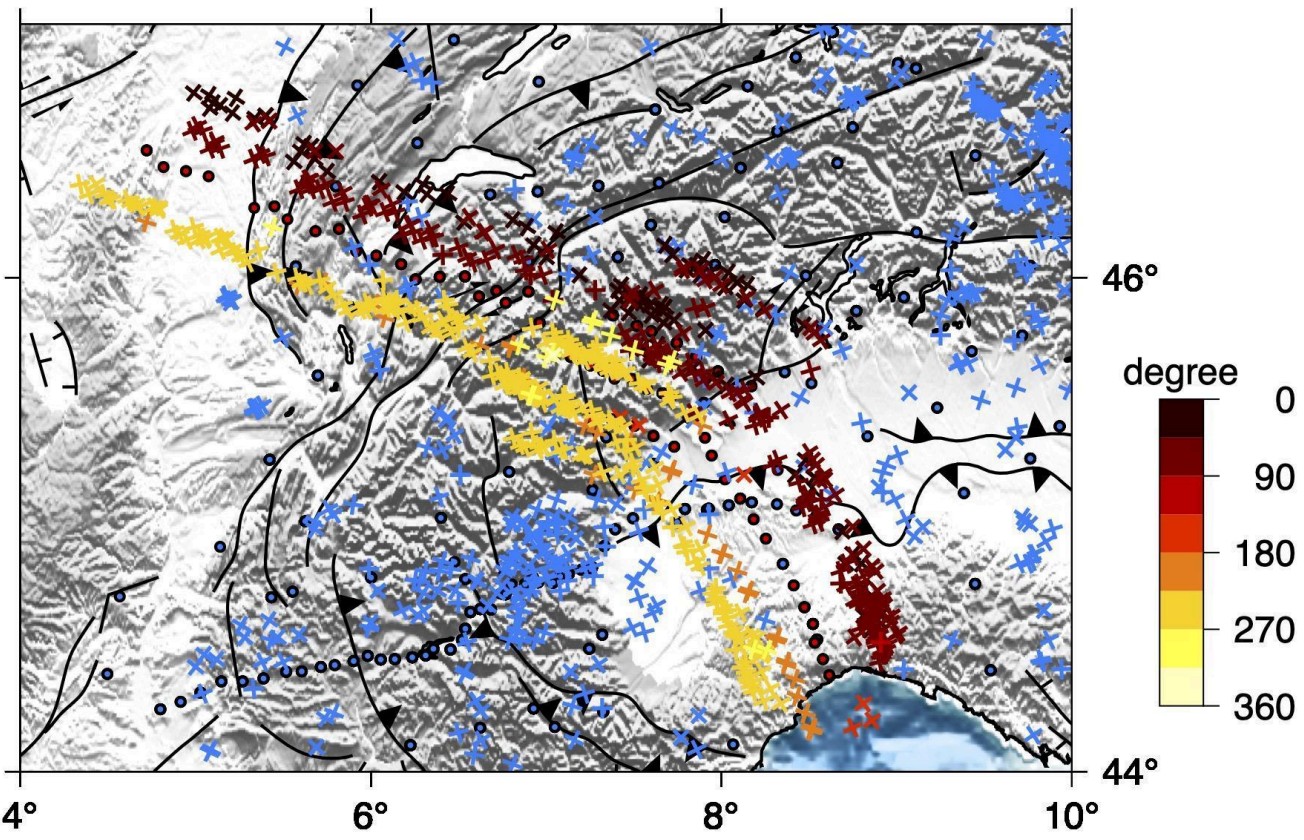

Figure S3 – Map of null measurements plotted at the piercing point of 150 km depth, marked with crosses rotated in the back azimuth direction and coloured following the color scale on the right, depending on the back azimuth;  in light blue all previous data.

Table S1 - Good single Splitting Intensity (SI) measurements. Header: Station | Station latitude | Station longitude | Earthquake ID (YYMMDDhhmm) | back-azimuth | SI |SI error

| Station | Lat | Lon | EQ ID | BAZ | SI | SI error |
|---|---|---|---|---|---|---|
| CE01 | 45.85 | 7.38 | 1809281002 | 73.70 | 0.56 | 0.04 |
| CE01 | 45.85 | 7.38 | 1812290339 | 64.00 | 0.06 | 0.09 |
| CE01 | 45.85 | 7.38 | 1901061727 | 66.54 | 0.65 | 0.04 |
| CE01 | 45.85 | 7.38 | 1901171506 | 52.65 | 0.75 | 0.26 |
| CE01 | 45.85 | 7.38 | 1901260812 | 66.06 | -0.16 | 0.18 |
| CE01 | 45.85 | 7.38 | 1903150503 | 245.84 | 0.39 | 0.15 |
| CE01 | 45.85 | 7.38 | 1904121140 | 72.66 | 0.44 | 0.07 |
| CE01 | 45.85 | 7.38 | 1904230537 | 61.60 | 0.43 | 0.06 |
| CE01 | 45.85 | 7.38 | 1905141258 | 46.67 | 1.24 | 0.09 |
| CE01 | 45.85 | 7.38 | 1905311012 | 64.11 | 0.31 | 0.13 |
| CE01 | 45.85 | 7.38 | 1907071508 | 68.18 | 0.24 | 0.23 |
| CE01 | 45.85 | 7.38 | 1907140943 | 67.84 | -0.40 | 0.38 |
| CE01 | 45.85 | 7.38 | 1909190706 | 83.83 | 0.62 | 0.06 |
| CE01 | 45.85 | 7.38 | 1909211953 | 69.62 | 0.01 | 0.17 |
| CE01 | 45.85 | 7.38 | 1909252346 | 69.16 | 0.32 | 0.17 |
| CE01 | 45.85 | 7.38 | 1910290104 | 64.96 | 1.83 | 0.16 |
| CE01 | 45.85 | 7.38 | 1911141617 | 67.24 | 1.11 | 0.06 |
| CE02 | 45.81 | 7.52 | 1809281025 | 74.30 | 1.92 | 0.28 |
| CE02 | 45.81 | 7.52 | 1901212359 | 81.62 | -0.54 | 0.23 |
| CE02 | 45.81 | 7.52 | 1902121234 | 40.73 | -0.08 | 0.28 |
| CE02 | 45.81 | 7.52 | 1904121140 | 72.78 | 0.74 | 0.36 |

| CE02 | 45.81 | 7.52 | 1906240253 | 70.77 | 0.85 | 0.03 |
|------|-------|------|------------|-------|------|------|
| CE02 | 45.81 | 7.52 | 1907140539 | 86.98 | 0.35 | 0.35 |
| CE02 | 45.81 | 7.52 | 1907140943 | 67.97 | -0.62 | 0.27 |
| CE02 | 45.81 | 7.52 | 1908021203 | 89.90 | 0.01 | 0.12 |
| CE02 | 45.81 | 7.52 | 1909190706 | 83.94 | 1.09 | 0.06 |
| CE02 | 45.81 | 7.52 | 1910290104 | 65.08 | 1.86 | 0.18 |
| CE03 | 45.79 | 7.60 | 1809281002 | 73.89 | 0.44 | 0.04 |
| CE03 | 45.79 | 7.60 | 1812290339 | 64.19 | 0.65 | 0.03 |
| CE03 | 45.79 | 7.60 | 1901181640 | 290.91 | -0.85 | 0.17 |
| CE03 | 45.79 | 7.60 | 1902020927 | 90.21 | 0.02 | 0.18 |
| CE03 | 45.79 | 7.60 | 1902021059 | 90.19 | 0.02 | 0.22 |
| CE03 | 45.79 | 7.60 | 1902021101 | 89.97 | 0.02 | 0.20 |
| CE03 | 45.79 | 7.60 | 1903150503 | 246.00 | -0.95 | 0.19 |
| CE03 | 45.79 | 7.60 | 1904121140 | 72.86 | 0.65 | 0.23 |
| CE03 | 45.79 | 7.60 | 1904230537 | 61.78 | 0.51 | 0.07 |
| CE03 | 45.79 | 7.60 | 1905062119 | 55.46 | 1.18 | 0.17 |
| CE03 | 45.79 | 7.60 | 1905311012 | 64.30 | 0.04 | 0.08 |
| CE03 | 45.79 | 7.60 | 1906140019 | 241.68 | 0.82 | 0.17 |
| CE03 | 45.79 | 7.60 | 1906240253 | 70.85 | 0.49 | 0.06 |
| CE03 | 45.79 | 7.60 | 1907071508 | 68.38 | 0.53 | 0.04 |
| CE03 | 45.79 | 7.60 | 1909141621 | 67.50 | -0.01 | 0.18 |
| CE03 | 45.79 | 7.60 | 1909290202 | 64.64 | 0.81 | 0.09 |
| CE03 | 45.79 | 7.60 | 1910161137 | 65.18 | 0.02 | 0.13 |
| CE03 | 45.79 | 7.60 | 1911141617 | 67.44 | 0.54 | 0.06 |
| CE04 | 45.74 | 7.75 | 1809281002 | 74.01 | 0.90 | 0.08 |

| CE04 | 45.74 | 7.75 | 1812290339 | 64.32 | 0.49 | 0.04 |
|---|---|---|---|---|---|---|
| CE04 | 45.74 | 7.75 | 1901061727 | 66.87 | 1.22 | 0.13 |
| CE04 | 45.74 | 7.75 | 1904220911 | 63.38 | -0.06 | 0.13 |
| CE04 | 45.74 | 7.75 | 1904230537 | 61.90 | 0.66 | 0.04 |
| CE04 | 45.74 | 7.75 | 1905311012 | 64.42 | 0.62 | 0.04 |
| CE04 | 45.74 | 7.75 | 1906240253 | 70.99 | 0.49 | 0.03 |
| CE04 | 45.74 | 7.75 | 1907071508 | 68.51 | 0.82 | 0.08 |
| CE04 | 45.74 | 7.75 | 1909190706 | 84.12 | 0.48 | 0.06 |
| CE04 | 45.74 | 7.75 | 1909290202 | 64.76 | 1.31 | 0.12 |
| CE04 | 45.74 | 7.75 | 1910290104 | 65.27 | 0.73 | 0.08 |
| CE04 | 45.74 | 7.75 | 1910310111 | 65.04 | -0.39 | 0.24 |
| CE04 | 45.74 | 7.75 | 1911052052 | 189.30 | 0.79 | 0.21 |
| CE04 | 45.74 | 7.75 | 1911150117 | 67.63 | 1.83 | 0.26 |
| CE05 | 45.68 | 7.86 | 1810292017 | 216.72 | 0.37 | 0.12 |
| CE05 | 45.68 | 7.86 | 1811040755 | 65.60 | 0.10 | 0.18 |
| CE05 | 45.68 | 7.86 | 1812290339 | 64.42 | 0.34 | 0.05 |
| CE05 | 45.68 | 7.86 | 1902081155 | 62.14 | 0.41 | 0.21 |
| CE05 | 45.68 | 7.86 | 1903240437 | 67.67 | 0.50 | 0.31 |
| CE05 | 45.68 | 7.86 | 1904121140 | 73.09 | 1.03 | 0.06 |
| CE05 | 45.68 | 7.86 | 1904230537 | 61.99 | 1.00 | 0.05 |
| CE05 | 45.68 | 7.86 | 1905311012 | 64.52 | 0.77 | 0.27 |
| CE05 | 45.68 | 7.86 | 1906240253 | 71.10 | 0.75 | 0.04 |
| CE05 | 45.68 | 7.86 | 1907071508 | 68.61 | 1.37 | 0.08 |
| CE05 | 45.68 | 7.86 | 1909290202 | 64.86 | 1.55 | 0.14 |
| CE05 | 45.68 | 7.86 | 1910310111 | 65.14 | 1.42 | 0.09 |

| CE05 | 45.68 | 7.86 | 1911141617 | 67.68 | 0.28 | 0.15 |
|------|-------|------|------------|-------|------|------|
| CE06 | 45.63 | 7.98 | 1812290339 | 64.54 | 0.58 | 0.08 |
| CE06 | 45.63 | 7.98 | 1901171506 | 53.37 | 1.42 | 0.22 |
| CE06 | 45.63 | 7.98 | 1903010850 | 251.39 | 1.21 | 0.10 |
| CE06 | 45.63 | 7.98 | 1904062155 | 74.88 | -0.16 | 0.07 |
| CE06 | 45.63 | 7.98 | 1904121140 | 73.20 | 0.68 | 0.06 |
| CE06 | 45.63 | 7.98 | 1904230537 | 62.10 | 0.66 | 0.08 |
| CE06 | 45.63 | 7.98 | 1906240253 | 71.23 | 1.27 | 0.03 |
| CE06 | 45.63 | 7.98 | 1907011659 | 64.54 | 1.01 | 0.20 |
| CE06 | 45.63 | 7.98 | 1907071508 | 68.73 | 1.48 | 0.05 |
| CE06 | 45.63 | 7.98 | 1907081852 | 68.65 | 0.96 | 0.09 |
| CE06 | 45.63 | 7.98 | 1907140943 | 68.41 | 1.43 | 0.37 |
| CE06 | 45.63 | 7.98 | 1907141026 | 68.08 | -1.18 | 0.07 |
| CE06 | 45.63 | 7.98 | 1911141617 | 67.79 | 1.26 | 0.05 |
| CE07 | 45.57 | 8.17 | 1809281002 | 74.38 | 0.77 | 0.11 |
| CE07 | 45.57 | 8.17 | 1812110226 | 198.19 | -0.23 | 0.06 |
| CE07 | 45.57 | 8.17 | 1812290339 | 64.70 | 0.95 | 0.08 |
| CE07 | 45.57 | 8.17 | 1904050956 | 41.09 | 0.61 | 0.26 |
| CE07 | 45.57 | 8.17 | 1904230537 | 62.25 | 0.35 | 0.07 |
| CE07 | 45.57 | 8.17 | 1906040439 | 41.36 | -0.27 | 0.10 |
| CE07 | 45.57 | 8.17 | 1907011659 | 64.70 | 0.20 | 0.44 |
| CE07 | 45.57 | 8.17 | 1907071508 | 68.91 | 0.78 | 0.09 |
| CE07 | 45.57 | 8.17 | 1907141026 | 68.25 | 1.83 | 0.13 |
| CI01 | 46.50 | 4.72 | 1809281014 | 71.35 | -1.19 | 0.12 |
| CI01 | 46.50 | 4.72 | 1901221901 | 153.32 | 0.65 | 0.11 |

| CI01 | 46.50 | 4.72 | 1902020927 | 88.08 | 0.90 | 0.14 |
|------|-------|------|------------|--------|-------|------|
| CI02 | 46.44 | 4.82 | 1809181157 | 40.67 | 0.00 | 0.09 |
| CI02 | 46.44 | 4.82 | 1809281014 | 71.45 | -1.35 | 0.08 |
| CI02 | 46.44 | 4.82 | 1810260905 | 37.16 | 1.27 | 0.46 |
| CI02 | 46.44 | 4.82 | 1812011327 | 69.30 | 0.00 | 0.05 |
| CI02 | 46.44 | 4.82 | 1812192137 | 47.80 | 0.46 | 0.12 |
| CI02 | 46.44 | 4.82 | 1901221901 | 153.39 | 1.11 | 0.09 |
| CI03 | 46.42 | 4.95 | 1809281002 | 71.64 | 0.28 | 0.04 |
| CI03 | 46.42 | 4.95 | 1809281014 | 71.55 | -1.57 | 0.22 |
| CI03 | 46.42 | 4.95 | 1809281025 | 72.11 | -1.37 | 0.18 |
| CI03 | 46.42 | 4.95 | 1904091753 | 196.05 | 1.38 | 0.07 |
| CI03 | 46.42 | 4.95 | 1904220911 | 61.24 | -0.06 | 0.12 |
| CI04 | 46.40 | 5.07 | 1901221901 | 153.56 | -0.52 | 0.16 |
| CI04 | 46.40 | 5.07 | 1902121234 | 38.50 | 1.47 | 0.24 |
| CI05 | 46.28 | 5.33 | 1809281025 | 72.46 | 1.21 | 0.16 |
| CI05 | 46.28 | 5.33 | 1903240437 | 65.43 | 0.03 | 0.12 |
| CI05 | 46.28 | 5.33 | 1908272355 | 196.24 | 0.27 | 0.08 |
| CI05 | 46.28 | 5.33 | 1910290242 | 63.18 | -0.02 | 0.11 |
| CI05 | 46.28 | 5.33 | 1911231211 | 60.00 | 0.24 | 0.13 |
| CI06 | 46.28 | 5.45 | 1901212359 | 79.84 | 0.40 | 0.08 |
| CI06 | 46.28 | 5.45 | 1906040439 | 39.28 | 0.34 | 0.05 |
| CI06 | 46.28 | 5.45 | 1907141026 | 65.75 | -0.92 | 0.42 |
| CI06 | 46.28 | 5.45 | 1911042153 | 238.70 | 0.03 | 0.15 |
| CI07 | 46.23 | 5.52 | 1810020016 | 79.26 | -0.42 | 0.17 |
| CI07 | 46.23 | 5.52 | 1810292326 | 282.59 | -0.02 | 0.20 |

| CI07 | 46.23 | 5.52 | 1812160942 | 57.14 | 0.55 | 0.10 |
|------|-------|------|------------|-------|------|------|
| CI07 | 46.23 | 5.52 | 1901181640 | 289.41 | 0.00 | 0.04 |
| CI07 | 46.23 | 5.52 | 1901200132 | 240.02 | 0.03 | 0.33 |
| CI07 | 46.23 | 5.52 | 1901212359 | 79.92 | 0.01 | 0.18 |
| CI07 | 46.23 | 5.52 | 1901220510 | 80.09 | 0.01 | 0.18 |
| CI07 | 46.23 | 5.52 | 1901221901 | 153.87 | 0.00 | 0.06 |
| CI07 | 46.23 | 5.52 | 1901260812 | 64.23 | 0.03 | 0.34 |
| CI07 | 46.23 | 5.52 | 1901301531 | 282.22 | 0.01 | 0.06 |
| CI07 | 46.23 | 5.52 | 1902020927 | 88.69 | 0.01 | 0.13 |
| CI07 | 46.23 | 5.52 | 1902021059 | 88.67 | 0.01 | 0.13 |
| CI07 | 46.23 | 5.52 | 1902021101 | 88.44 | 0.01 | 0.13 |
| CI07 | 46.23 | 5.52 | 1902081155 | 60.18 | 0.04 | 0.43 |
| CI07 | 46.23 | 5.52 | 1904062155 | 72.64 | 0.20 | 0.05 |
| CI07 | 46.23 | 5.52 | 1907141026 | 65.84 | -0.70 | 0.24 |
| CI07 | 46.23 | 5.52 | 1911042153 | 238.74 | 1.88 | 0.45 |
| CI08 | 46.18 | 5.68 | 1810260905 | 38.00 | 0.47 | 0.08 |
| CI08 | 46.18 | 5.68 | 1811011930 | 196.58 | 1.32 | 0.12 |
| CI08 | 46.18 | 5.68 | 1901220510 | 80.24 | -0.66 | 0.08 |
| CI08 | 46.18 | 5.68 | 1901260351 | 42.14 | 0.83 | 0.30 |
| CI08 | 46.18 | 5.68 | 1907141026 | 65.99 | -0.54 | 0.29 |
| CI08 | 46.18 | 5.68 | 1909290202 | 62.99 | 1.21 | 0.32 |
| CI08 | 46.18 | 5.68 | 1911161019 | 65.86 | 0.26 | 0.19 |
| CI09 | 46.19 | 5.82 | 1809230552 | 42.15 | -0.47 | 0.19 |
| CI09 | 46.19 | 5.82 | 1812011327 | 70.27 | -0.60 | 0.10 |
| CI09 | 46.19 | 5.82 | 1901220510 | 80.34 | -0.85 | 0.08 |

| CI09 | 46.19 | 5.82 | 1903060013 | 60.97 | 0.69 | 0.13 |
|------|-------|------|------------|--------|-------|------|
| CI09 | 46.19 | 5.82 | 1904051614 | 198.90 | -0.01 | 0.12 |
| CI09 | 46.19 | 5.82 | 1911141845 | 66.04 | -1.48 | 0.33 |
| CI10 | 46.11 | 5.92 | 1809281002 | 72.50 | 0.17 | 0.04 |
| CI10 | 46.11 | 5.92 | 1812110226 | 197.10 | 1.31 | 0.07 |
| CI10 | 46.11 | 5.92 | 1812280303 | 61.53 | 0.21 | 0.15 |
| CI10 | 46.11 | 5.92 | 1901200132 | 240.25 | 0.02 | 0.25 |
| CI10 | 46.11 | 5.92 | 1909190732 | 82.63 | -0.44 | 0.04 |
| CI11 | 46.09 | 6.03 | 1809181157 | 42.41 | 0.02 | 0.10 |
| CI11 | 46.09 | 6.03 | 1810292326 | 282.94 | 0.24 | 0.31 |
| CI11 | 46.09 | 6.03 | 1811011930 | 196.75 | 0.75 | 0.30 |
| CI11 | 46.09 | 6.03 | 1812110226 | 197.16 | 1.04 | 0.09 |
| CI11 | 46.09 | 6.03 | 1901212359 | 80.37 | 0.09 | 0.07 |
| CI11 | 46.09 | 6.03 | 1902021059 | 89.05 | -0.90 | 0.19 |
| CI11 | 46.09 | 6.03 | 1905162252 | 282.79 | 0.54 | 0.17 |
| CI11 | 46.09 | 6.03 | 1906040439 | 39.73 | 0.68 | 0.24 |
| CI11 | 46.09 | 6.03 | 1907071508 | 67.01 | 0.69 | 0.03 |
| CI12 | 46.05 | 6.17 | 1812110226 | 197.23 | 1.06 | 0.11 |
| CI12 | 46.05 | 6.17 | 1812290339 | 62.97 | 0.03 | 0.04 |
| CI12 | 46.05 | 6.17 | 1901260351 | 42.81 | 0.93 | 0.17 |
| CI12 | 46.05 | 6.17 | 1905162252 | 282.89 | 1.74 | 0.33 |
| CI12 | 46.05 | 6.17 | 1906040439 | 39.84 | 0.67 | 0.07 |
| CI12 | 46.05 | 6.17 | 1906240253 | 69.53 | 0.50 | 0.05 |
| CI12 | 46.05 | 6.17 | 1907071508 | 67.13 | 0.70 | 0.03 |
| CI12 | 46.05 | 6.17 | 1910290104 | 63.96 | 0.44 | 0.06 |

| CI12 | 46.05 | 6.17 | 1910310111 | 63.72 | 1.00 | 0.13 |
|------|-------|------|------------|-------|------|------|
| CI13 | 45.99 | 6.26 | 1809281014 | 72.70 | 1.80 | 0.17 |
| CI13 | 45.99 | 6.26 | 1810102200 | 46.90 | -0.34 | 0.24 |
| CI13 | 45.99 | 6.26 | 1812290339 | 63.06 | 0.01 | 0.09 |
| CI13 | 45.99 | 6.26 | 1901260351 | 42.95 | 0.80 | 0.10 |
| CI13 | 45.99 | 6.26 | 1906040439 | 39.90 | 1.02 | 0.08 |
| CI14 | 46.00 | 6.39 | 1809281025 | 73.37 | 1.72 | 0.49 |
| CI14 | 46.00 | 6.39 | 1811040755 | 64.40 | 0.89 | 0.18 |
| CI14 | 46.00 | 6.39 | 1901061727 | 65.69 | 1.04 | 0.11 |
| CI14 | 46.00 | 6.39 | 1901221901 | 154.45 | -0.78 | 0.08 |
| CI14 | 46.00 | 6.39 | 1904051614 | 199.19 | -0.05 | 0.21 |
| CI14 | 46.00 | 6.39 | 1904221449 | 198.79 | 0.10 | 0.17 |
| CI14 | 46.00 | 6.39 | 1911141845 | 66.57 | -0.04 | 0.20 |
| CI15 | 46.01 | 6.54 | 1809281002 | 73.01 | 0.35 | 0.03 |
| CI15 | 46.01 | 6.54 | 1810102200 | 47.20 | -0.91 | 0.21 |
| CI15 | 46.01 | 6.54 | 1810290654 | 216.44 | 0.07 | 0.15 |
| CI15 | 46.01 | 6.54 | 1812290339 | 63.28 | 0.02 | 0.08 |
| CI15 | 46.01 | 6.54 | 1901260812 | 65.24 | 0.46 | 0.12 |
| CI15 | 46.01 | 6.54 | 1902081155 | 61.04 | -0.48 | 0.21 |
| CI15 | 46.01 | 6.54 | 1905062119 | 54.23 | 1.04 | 0.07 |
| CI15 | 46.01 | 6.54 | 1905162252 | 283.15 | 0.81 | 0.20 |
| CI15 | 46.01 | 6.54 | 1906191724 | 58.74 | 0.45 | 0.09 |
| CI15 | 46.01 | 6.54 | 1906281551 | 40.68 | 1.15 | 0.03 |
| CI15 | 46.01 | 6.54 | 1907071508 | 67.44 | 0.68 | 0.06 |
| CI15 | 46.01 | 6.54 | 1908272355 | 196.79 | 0.85 | 0.03 |

| CI15 | 46.01 | 6.54 | 1911141617 | 66.51 | 0.80 | 0.04 |
|------|-------|------|------------|--------|-------|------|
| CI16 | 45.92 | 6.62 | 1809281002 | 73.09 | 0.62 | 0.04 |
| CI16 | 45.92 | 6.62 | 1809281025 | 73.57 | -0.49 | 0.16 |
| CI16 | 45.92 | 6.62 | 1812190137 | 253.14 | -0.03 | 0.25 |
| CI16 | 45.92 | 6.62 | 1901221901 | 154.61 | -0.01 | 0.06 |
| CI16 | 45.92 | 6.62 | 1901260812 | 65.35 | 0.03 | 0.32 |
| CI16 | 45.92 | 6.62 | 1901301531 | 282.96 | 0.01 | 0.06 |
| CI16 | 45.92 | 6.62 | 1902020927 | 89.50 | 0.01 | 0.13 |
| CI16 | 45.92 | 6.62 | 1902021059 | 89.48 | 0.01 | 0.13 |
| CI16 | 45.92 | 6.62 | 1902021101 | 89.26 | 0.01 | 0.13 |
| CI16 | 45.92 | 6.62 | 1902081155 | 61.11 | 0.05 | 0.40 |
| CI16 | 45.92 | 6.62 | 1903081506 | 61.23 | 1.55 | 0.19 |
| CI16 | 45.92 | 6.62 | 1904230537 | 60.99 | 1.01 | 0.05 |
| CI16 | 45.92 | 6.62 | 1905062119 | 54.37 | 0.99 | 0.07 |
| CI16 | 45.92 | 6.62 | 1905141258 | 45.78 | 0.87 | 0.19 |
| CI16 | 45.92 | 6.62 | 1906040439 | 40.18 | 0.97 | 0.07 |
| CI16 | 45.92 | 6.62 | 1906191724 | 58.85 | 1.38 | 0.11 |
| CI16 | 45.92 | 6.62 | 1906240253 | 69.96 | 0.59 | 0.06 |
| CI16 | 45.92 | 6.62 | 1906281551 | 40.76 | 1.33 | 0.05 |
| CI16 | 45.92 | 6.62 | 1907071508 | 67.54 | 0.87 | 0.04 |
| CI16 | 45.92 | 6.62 | 1907140910 | 66.78 | 1.30 | 0.11 |
| CI16 | 45.92 | 6.62 | 1907141026 | 66.86 | 1.09 | 0.21 |
| CI16 | 45.92 | 6.62 | 1908272355 | 196.83 | 1.32 | 0.05 |
| CI16 | 45.92 | 6.62 | 1910290844 | 44.71 | -0.28 | 0.10 |
| CI17 | 45.95 | 6.72 | 1809281002 | 73.16 | 0.45 | 0.07 |

| | | | | | | |
|---|---|---|---|---|---|---|
| CI17 | 45.95 | 6.72 | 1810020016 | 80.31 | 1.45 | 0.23 |
| CI17 | 45.95 | 6.72 | 1810101844 | 82.37 | -0.09 | 0.11 |
| CI17 | 45.95 | 6.72 | 1812290339 | 63.44 | 0.03 | 0.14 |
| CI17 | 45.95 | 6.72 | 1901181640 | 290.27 | -0.68 | 0.13 |
| CI17 | 45.95 | 6.72 | 1901200132 | 240.73 | 0.38 | 0.11 |
| CI17 | 45.95 | 6.72 | 1901221901 | 154.67 | -0.92 | 0.10 |
| CI17 | 45.95 | 6.72 | 1901301531 | 283.04 | 0.80 | 0.11 |
| CI17 | 45.95 | 6.72 | 1902121234 | 40.02 | 0.45 | 0.26 |
| CI17 | 45.95 | 6.72 | 1903041006 | 53.47 | 1.24 | 0.44 |
| CI17 | 45.95 | 6.72 | 1903081506 | 61.30 | 0.65 | 0.09 |
| CI17 | 45.95 | 6.72 | 1904091753 | 196.91 | 0.41 | 0.10 |
| CI17 | 45.95 | 6.72 | 1906040439 | 40.25 | 0.73 | 0.07 |
| CI17 | 45.95 | 6.72 | 1906281551 | 40.85 | 0.82 | 0.03 |
| CI17 | 45.95 | 6.72 | 1907071508 | 67.61 | 0.70 | 0.03 |
| CI17 | 45.95 | 6.72 | 1909190706 | 83.31 | 0.84 | 0.05 |
| CI17 | 45.95 | 6.72 | 1910161137 | 64.44 | 0.37 | 0.08 |
| CI18 | 45.90 | 6.77 | 1809281002 | 73.22 | 0.54 | 0.07 |
| CI18 | 45.90 | 6.77 | 1810012359 | 80.36 | 0.55 | 0.15 |
| CI18 | 45.90 | 6.77 | 1812110226 | 197.51 | 0.19 | 0.08 |
| CI18 | 45.90 | 6.77 | 1901221901 | 154.71 | -1.29 | 0.14 |
| CI18 | 45.90 | 6.77 | 1901260351 | 43.62 | -0.01 | 0.15 |
| CI18 | 45.90 | 6.77 | 1901260812 | 65.50 | -0.03 | 0.35 |
| CI18 | 45.90 | 6.77 | 1901301531 | 283.07 | 0.03 | 0.47 |
| CI18 | 45.90 | 6.77 | 1902121234 | 40.07 | 0.74 | 0.17 |
| CI18 | 45.90 | 6.77 | 1902171435 | 46.02 | 0.20 | 0.31 |

| CI18 | 45.90 | 6.77 | 1906191724 | 59.00 | 0.42 | 0.09 |
|------|-------|------|------------|-------|------|------|
| CI18 | 45.90 | 6.77 | 1907140910 | 66.91 | 1.34 | 0.18 |
| CI19 | 45.94 | 6.90 | 1812192137 | 50.38 | -0.86 | 0.19 |
| CI19 | 45.94 | 6.90 | 1903010850 | 250.64 | 0.52 | 0.05 |
| CI19 | 45.94 | 6.90 | 1904051614 | 199.46 | 1.10 | 0.08 |
| CI19 | 45.94 | 6.90 | 1904062155 | 73.88 | 1.87 | 0.14 |
| CI19 | 45.94 | 6.90 | 1904091753 | 197.00 | 0.26 | 0.20 |
| CI19 | 45.94 | 6.90 | 1904121140 | 72.25 | 0.77 | 0.06 |
| CI19 | 45.94 | 6.90 | 1906040439 | 40.39 | 1.24 | 0.13 |
| CI19 | 45.94 | 6.90 | 1906191724 | 59.10 | 1.39 | 0.08 |
| CI19 | 45.94 | 6.90 | 1907071508 | 67.76 | 0.94 | 0.04 |
| CI19 | 45.94 | 6.90 | 1908272355 | 196.96 | 0.72 | 0.04 |
| CI19 | 45.94 | 6.90 | 1909291557 | 237.91 | 0.79 | 0.07 |
| CI19 | 45.94 | 6.90 | 1911141617 | 66.82 | 1.58 | 0.05 |
| CI21 | 45.72 | 7.14 | 1809280659 | 73.71 | 0.41 | 0.34 |
| CI21 | 45.72 | 7.14 | 1809281002 | 73.55 | 1.08 | 0.10 |
| CI21 | 45.72 | 7.14 | 1809281014 | 73.46 | 1.54 | 0.09 |
| CI21 | 45.72 | 7.14 | 1810141241 | 126.72 | 0.04 | 0.17 |
| CI21 | 45.72 | 7.14 | 1812290339 | 63.84 | 0.84 | 0.08 |
| CI21 | 45.72 | 7.14 | 1901181640 | 290.55 | 0.07 | 0.38 |
| CI21 | 45.72 | 7.14 | 1901212359 | 81.37 | 0.15 | 0.23 |
| CI21 | 45.72 | 7.14 | 1901220510 | 81.54 | 0.26 | 0.19 |
| CI21 | 45.72 | 7.14 | 1901260351 | 44.17 | 0.94 | 0.26 |
| CI21 | 45.72 | 7.14 | 1906240253 | 70.49 | 0.97 | 0.04 |
| CI21 | 45.72 | 7.14 | 1907071508 | 68.03 | 1.11 | 0.05 |

| CI21 | 45.72 | 7.14 | 1907140539 | 86.75 | 1.16 | 0.12 |
|------|-------|------|------------|-------|------|------|
| CI21 | 45.72 | 7.14 | 1909141621 | 67.14 | -0.09 | 0.16 |
| CI21 | 45.72 | 7.14 | 1909190732 | 83.64 | 1.50 | 0.26 |
| CI21 | 45.72 | 7.14 | 1911150117 | 67.15 | -1.21 | 0.14 |
| CI22 | 45.68 | 7.14 | 1809281002 | 73.56 | 0.80 | 0.07 |
| CI22 | 45.68 | 7.14 | 1809281014 | 73.47 | 1.94 | 0.20 |
| CI22 | 45.68 | 7.14 | 1905162252 | 283.51 | -0.04 | 0.16 |
| CI22 | 45.68 | 7.14 | 1907011659 | 63.88 | 1.96 | 0.14 |
| CI22 | 45.68 | 7.14 | 1907071508 | 68.04 | 0.97 | 0.09 |
| CI22 | 45.68 | 7.14 | 1907140539 | 86.77 | 0.92 | 0.16 |
| CI23 | 45.68 | 7.24 | 1809281002 | 73.64 | 1.07 | 0.07 |
| CI23 | 45.68 | 7.24 | 1810020016 | 80.83 | -0.37 | 0.17 |
| CI23 | 45.68 | 7.24 | 1810141241 | 126.79 | -1.21 | 0.16 |
| CI23 | 45.68 | 7.24 | 1812160942 | 59.06 | -0.01 | 0.06 |
| CI23 | 45.68 | 7.24 | 1812161426 | 96.66 | -0.02 | 0.25 |
| CI23 | 45.68 | 7.24 | 1812290339 | 63.93 | 0.39 | 0.06 |
| CI23 | 45.68 | 7.24 | 1901061727 | 66.48 | 0.46 | 0.10 |
| CI23 | 45.68 | 7.24 | 1901181640 | 290.62 | -0.91 | 0.21 |
| CI23 | 45.68 | 7.24 | 1903060013 | 62.22 | 0.48 | 0.08 |
| CI23 | 45.68 | 7.24 | 1904051614 | 199.61 | 0.04 | 0.14 |
| CI23 | 45.68 | 7.24 | 1904230537 | 61.52 | 0.86 | 0.06 |
| CI23 | 45.68 | 7.24 | 1905030725 | 39.67 | 1.39 | 0.16 |
| CI23 | 45.68 | 7.24 | 1905311012 | 64.03 | 0.60 | 0.07 |
| CI23 | 45.68 | 7.24 | 1906240253 | 70.60 | 0.95 | 0.04 |
| CI23 | 45.68 | 7.24 | 1907071508 | 68.12 | 0.90 | 0.05 |

| CI23 | 45.68 | 7.24 | 1910290104 | 64.89 | 0.79 | 0.08 |
|------|-------|------|------------|-------|------|------|
| CI23 | 45.68 | 7.24 | 1911141617 | 67.18 | 1.19 | 0.13 |
| CI24 | 45.65 | 7.25 | 1809281002 | 73.66 | 1.07 | 0.09 |
| CI24 | 45.65 | 7.25 | 1812290339 | 63.95 | 0.31 | 0.05 |
| CI24 | 45.65 | 7.25 | 1902021101 | 89.73 | -0.33 | 0.24 |
| CI24 | 45.65 | 7.25 | 1904062155 | 74.29 | 0.19 | 0.09 |
| CI24 | 45.65 | 7.25 | 1904230537 | 61.53 | 0.86 | 0.05 |
| CI24 | 45.65 | 7.25 | 1905062119 | 55.19 | 0.07 | 0.30 |
| CI24 | 45.65 | 7.25 | 1906240253 | 70.62 | 1.03 | 0.05 |
| CI24 | 45.65 | 7.25 | 1907071508 | 68.14 | 0.90 | 0.05 |
| CI24 | 45.65 | 7.25 | 1910142223 | 90.32 | 1.03 | 0.06 |
| CI24 | 45.65 | 7.25 | 1910290104 | 64.90 | 0.92 | 0.06 |
| CI24 | 45.65 | 7.25 | 1911052052 | 189.03 | -0.24 | 0.25 |
| CI24 | 45.65 | 7.25 | 1911231211 | 61.89 | 0.55 | 0.07 |
| CI25 | 45.63 | 7.32 | 1809281002 | 73.72 | 0.97 | 0.07 |
| CI25 | 45.63 | 7.32 | 1809281025 | 74.20 | 0.64 | 0.14 |
| CI25 | 45.63 | 7.32 | 1812290339 | 64.01 | 0.44 | 0.04 |
| CI25 | 45.63 | 7.32 | 1901220510 | 81.72 | 0.98 | 0.08 |
| CI25 | 45.63 | 7.32 | 1902121234 | 40.60 | -0.19 | 0.13 |
| CI25 | 45.63 | 7.32 | 1903041006 | 54.27 | -0.55 | 0.14 |
| CI25 | 45.63 | 7.32 | 1904230537 | 61.59 | 0.57 | 0.05 |
| CI25 | 45.63 | 7.32 | 1905311012 | 64.11 | -1.80 | 0.38 |
| CI25 | 45.63 | 7.32 | 1906240253 | 70.69 | 1.04 | 0.06 |
| CI25 | 45.63 | 7.32 | 1907071508 | 68.21 | 0.91 | 0.10 |
| CI25 | 45.63 | 7.32 | 1907141026 | 67.54 | -0.92 | 0.19 |

| CI25 | 45.63 | 7.32 | 1910290104 | 64.96 | 1.17 | 0.13 |
|------|-------|------|------------|-------|------|------|
| CI25 | 45.63 | 7.32 | 1911141617 | 67.27 | 0.27 | 0.08 |
| CI26 | 45.60 | 7.39 | 1901220510 | 81.78 | 1.15 | 0.21 |
| CI26 | 45.60 | 7.39 | 1903101248 | 51.72 | 0.71 | 0.28 |
| CI26 | 45.60 | 7.39 | 1907071508 | 68.27 | 1.17 | 0.08 |
| CI26 | 45.60 | 7.39 | 1907141026 | 67.61 | 1.75 | 0.45 |
| CI27 | 45.60 | 7.49 | 1810020016 | 81.05 | 0.39 | 0.48 |
| CI28 | 45.56 | 7.57 | 1904230537 | 61.79 | 0.45 | 0.11 |
| CI28 | 45.56 | 7.57 | 1907140943 | 68.10 | 1.42 | 0.38 |
| CI29 | 45.54 | 7.66 | 1905311012 | 64.41 | 0.44 | 0.24 |
| CI29 | 45.54 | 7.66 | 1906240253 | 71.01 | 1.09 | 0.15 |
| CI29 | 45.54 | 7.66 | 1907071508 | 68.51 | 0.79 | 0.39 |
| CI29 | 45.54 | 7.66 | 1905311012 | 64.41 | 0.44 | 0.24 |
| CI29 | 45.54 | 7.66 | 1906240253 | 71.01 | 1.09 | 0.15 |
| CI29 | 45.54 | 7.66 | 1907071508 | 68.51 | 0.79 | 0.39 |
| CI30 | 45.52 | 7.71 | 1812192137 | 51.51 | -1.19 | 0.25 |
| CI30 | 45.52 | 7.71 | 1902121234 | 40.95 | -1.45 | 0.16 |
| CI30 | 45.52 | 7.71 | 1907071508 | 68.55 | 1.23 | 0.11 |
| CI30 | 45.52 | 7.71 | 1909141621 | 67.68 | 0.49 | 0.29 |
| CI31 | 45.44 | 7.81 | 1903010850 | 251.27 | 0.58 | 0.05 |
| CI31 | 45.44 | 7.81 | 1904220911 | 63.45 | 0.23 | 0.11 |
| CI31 | 45.44 | 7.81 | 1906040439 | 41.09 | -0.65 | 0.15 |
| CI31 | 45.44 | 7.81 | 1910161137 | 65.42 | 0.49 | 0.08 |
| CI32 | 45.36 | 7.91 | 1810102113 | 50.39 | -0.81 | 0.23 |
| CI32 | 45.36 | 7.91 | 1905311012 | 64.64 | 0.74 | 0.20 |

| | | | | | | |
|------|-------|------|------------|--------|-------|------|
| CI32 | 45.36 | 7.91 | 1907011659 | 64.53  | -1.09 | 0.26 |
| CI32 | 45.36 | 7.91 | 1910290104 | 65.48  | 0.46  | 0.14 |
| CI33 | 45.29 | 7.94 | 1812161426 | 97.32  | -0.93 | 0.12 |
| CI33 | 45.29 | 7.94 | 1901200132 | 241.36 | 0.65  | 0.14 |
| CI33 | 45.29 | 7.94 | 1902021101 | 90.25  | 1.85  | 0.19 |
| CI33 | 45.29 | 7.94 | 1902081155 | 62.28  | 0.27  | 0.09 |
| CI33 | 45.29 | 7.94 | 1906281551 | 41.99  | -1.47 | 0.07 |
| CI33 | 45.29 | 7.94 | 1907141026 | 68.17  | 0.08  | 0.40 |
| CI34 | 45.19 | 8.02 | 1810102045 | 50.26  | 1.44  | 0.13 |
| CI34 | 45.19 | 8.02 | 1901301531 | 283.79 | 0.19  | 0.18 |
| CI34 | 45.19 | 8.02 | 1906040439 | 41.26  | -0.86 | 0.13 |
| CI35 | 45.11 | 8.11 | 1811040755 | 65.91  | -0.89 | 0.11 |
| CI35 | 45.11 | 8.11 | 1911021808 | 199.25 | -0.63 | 0.14 |
| CI36 | 45.03 | 8.18 | 1810020016 | 81.81  | -1.17 | 0.24 |
| CI36 | 45.03 | 8.18 | 1901220510 | 82.60  | -0.71 | 0.07 |
| CI36 | 45.03 | 8.18 | 1902020927 | 90.67  | -0.09 | 0.23 |
| CI36 | 45.03 | 8.18 | 1904230537 | 62.34  | 0.38  | 0.07 |
| CI36 | 45.03 | 8.18 | 1905311012 | 64.94  | 0.82  | 0.12 |
| CI37 | 44.95 | 8.25 | 1812161426 | 97.69  | -1.46 | 0.41 |
| CI37 | 44.95 | 8.25 | 1902121234 | 41.51  | 0.00  | 0.13 |
| CI37 | 44.95 | 8.25 | 1903150503 | 246.43 | -0.63 | 0.14 |
| CI37 | 44.95 | 8.25 | 1905311012 | 65.01  | 0.37  | 0.18 |
| CI37 | 44.95 | 8.25 | 1907150821 | 52.80  | -0.92 | 0.11 |
| CI37 | 44.95 | 8.25 | 1909141621 | 68.34  | 0.77  | 0.30 |
| CI37 | 44.95 | 8.25 | 1910161137 | 65.87  | 0.04  | 0.09 |

| CI38 | 44.76 | 8.41 | 1810290654 | 216.76 | 0.00 | 0.02 |
|------|-------|------|------------|--------|-------|------|
| CI38 | 44.76 | 8.41 | 1810292017 | 216.62 | 0.00 | 0.02 |
| CI38 | 44.76 | 8.41 | 1810292326 | 284.31 | -0.02 | 0.16 |
| CI38 | 44.76 | 8.41 | 1903010850 | 251.66 | -0.51 | 0.05 |
| CI38 | 44.76 | 8.41 | 1904230537 | 62.56 | 0.04 | 0.11 |
| CI38 | 44.76 | 8.41 | 1905311012 | 65.18 | -1.31 | 0.20 |
| CI38 | 44.76 | 8.41 | 1906250601 | 127.59 | 0.01 | 0.06 |
| CI38 | 44.76 | 8.41 | 1906281551 | 42.47 | -0.01 | 0.09 |
| CI38 | 44.76 | 8.41 | 1907011659 | 65.02 | -0.02 | 0.24 |
| CI38 | 44.76 | 8.41 | 1907071508 | 69.35 | -0.02 | 0.24 |
| CI38 | 44.76 | 8.41 | 1907081852 | 69.27 | -0.02 | 0.23 |
| CI38 | 44.76 | 8.41 | 1907122042 | 63.50 | -0.02 | 0.18 |
| CI38 | 44.76 | 8.41 | 1907140910 | 68.66 | -0.02 | 0.23 |
| CI38 | 44.76 | 8.41 | 1907140943 | 69.06 | -0.02 | 0.23 |
| CI38 | 44.76 | 8.41 | 1907141026 | 68.73 | -0.02 | 0.23 |
| CI38 | 44.76 | 8.41 | 1907150821 | 53.09 | -0.01 | 0.12 |
| CI38 | 44.76 | 8.41 | 1910290104 | 66.01 | -0.06 | 0.37 |
| CI38 | 44.76 | 8.41 | 1910290242 | 65.95 | -0.03 | 0.37 |
| CI38 | 44.76 | 8.41 | 1910310111 | 65.77 | -0.02 | 0.36 |
| CI38 | 44.76 | 8.41 | 1911021808 | 199.38 | 0.00 | 0.04 |
| CI38 | 44.76 | 8.41 | 1911042153 | 240.19 | -0.02 | 0.21 |
| CI38 | 44.76 | 8.41 | 1911052052 | 189.62 | 0.00 | 0.03 |
| CI38 | 44.76 | 8.41 | 1911141845 | 68.58 | -0.03 | 0.42 |
| CI38 | 44.76 | 8.41 | 1911142112 | 68.46 | -0.38 | 0.41 |
| CI38 | 44.76 | 8.41 | 1911150117 | 68.46 | -0.03 | 0.43 |

| CI38 | 44.76 | 8.41 | 1911161019 | 68.50 | -0.02 | 0.42 |
|------|-------|------|------------|-------|-------|------|
| CI38 | 44.76 | 8.41 | 1911231211 | 63.20 | -0.03 | 0.31 |
| CI38 | 44.76 | 8.41 | 1912030846 | 249.30 | -0.03 | 0.34 |
| CI39 | 44.67 | 8.47 | 1810290654 | 216.75 | -1.12 | 0.15 |
| CI39 | 44.67 | 8.47 | 1810292326 | 284.33 | 1.74 | 0.24 |
| CI39 | 44.67 | 8.47 | 1902020927 | 90.90 | 0.09 | 0.30 |
| CI39 | 44.67 | 8.47 | 1907140943 | 69.14 | 1.87 | 0.44 |
| CI39 | 44.67 | 8.47 | 1908272355 | 197.59 | 0.92 | 0.03 |
| CI39 | 44.67 | 8.47 | 1911231211 | 63.28 | -0.20 | 0.15 |
| CI40 | 44.60 | 8.52 | 1902020927 | 90.94 | -1.97 | 0.25 |
| CI40 | 44.60 | 8.52 | 1904221449 | 199.78 | 0.30 | 0.38 |
| CI40 | 44.60 | 8.52 | 1906191724 | 61.22 | -0.75 | 0.07 |
| CI40 | 44.60 | 8.52 | 1907140539 | 88.35 | -0.78 | 0.23 |
| CI40 | 44.60 | 8.52 | 1909141621 | 68.68 | 0.27 | 0.22 |
| CI41 | 44.53 | 8.53 | 1903010850 | 251.73 | -1.10 | 0.07 |
| CI41 | 44.53 | 8.53 | 1904062155 | 75.78 | -0.43 | 0.06 |
| CI41 | 44.53 | 8.53 | 1906140019 | 241.98 | -0.57 | 0.04 |
| CI41 | 44.53 | 8.53 | 1906240253 | 72.19 | -0.45 | 0.08 |
| CI41 | 44.53 | 8.53 | 1906281551 | 42.61 | -0.73 | 0.04 |
| CI41 | 44.53 | 8.53 | 1908272355 | 197.60 | 0.78 | 0.02 |
| CI41 | 44.53 | 8.53 | 1909291557 | 238.46 | -0.77 | 0.04 |
| CI42 | 44.46 | 8.58 | 1901220510 | 83.12 | 1.14 | 0.23 |
| CI42 | 44.46 | 8.58 | 1901221901 | 155.96 | 0.87 | 0.05 |
| CI42 | 44.46 | 8.58 | 1903150503 | 246.65 | -1.35 | 0.20 |
| CI42 | 44.46 | 8.58 | 1906240253 | 72.26 | -0.60 | 0.10 |

| CI42 | 44.46 | 8.58 | 1909291557 | 238.47 | -1.21 | 0.07 |
|------|-------|------|------------|--------|-------|------|
| CI42 | 44.46 | 8.58 | 1911142112 | 68.69 | -0.49 | 0.05 |
| CI43 | 44.40 | 8.62 | 1811011930 | 197.91 | -0.01 | 0.12 |
| CI43 | 44.40 | 8.62 | 1811040755 | 66.44 | 0.03 | 0.27 |
| CI43 | 44.40 | 8.62 | 1901221901 | 155.98 | 0.95 | 0.07 |
| CI43 | 44.40 | 8.62 | 1902171435 | 48.88 | -0.70 | 0.10 |
| CI43 | 44.40 | 8.62 | 1903010850 | 251.79 | -0.70 | 0.09 |
| CI43 | 44.40 | 8.62 | 1906240253 | 72.32 | -0.32 | 0.06 |
| CI43 | 44.40 | 8.62 | 1908011828 | 238.95 | -0.88 | 0.07 |
| CI43 | 44.40 | 8.62 | 1909190706 | 85.08 | -1.42 | 0.09 |
| CI43 | 44.40 | 8.62 | 1909291557 | 238.47 | -1.08 | 0.08 |
| CW01 | 45.43 | 7.26 | 1810292017 | 216.40 | -0.22 | 0.40 |
| CW01 | 45.43 | 7.26 | 1811040755 | 65.19 | 1.35 | 0.32 |
| CW01 | 45.43 | 7.26 | 1812161426 | 96.79 | -0.72 | 0.17 |
| CW01 | 45.43 | 7.26 | 1901221901 | 155.05 | 0.39 | 0.12 |
| CW01 | 45.43 | 7.26 | 1903041006 | 54.32 | 0.19 | 0.27 |
| CW01 | 45.43 | 7.26 | 1903150503 | 245.75 | 1.23 | 0.15 |
| CW01 | 45.43 | 7.26 | 1907071508 | 68.22 | 0.89 | 0.05 |
| CW01 | 45.43 | 7.26 | 1907141026 | 67.56 | 0.47 | 0.10 |
| CW02 | 45.43 | 7.37 | 1810290654 | 216.58 | 1.77 | 0.12 |
| CW02 | 45.43 | 7.37 | 1810292017 | 216.45 | -0.22 | 0.10 |
| CW02 | 45.43 | 7.37 | 1901181640 | 290.69 | -1.46 | 0.48 |
| CW02 | 45.43 | 7.37 | 1902121234 | 40.67 | -0.78 | 0.12 |
| CW02 | 45.43 | 7.37 | 1906250601 | 126.80 | -1.91 | 0.19 |
| CW02 | 45.43 | 7.37 | 1907071508 | 68.31 | 1.07 | 0.09 |

| CW02 | 45.43 | 7.37 | 1908142135 | 302.99 | -0.04 | 0.36 |
|------|-------|------|------------|--------|-------|------|
| CW02 | 45.43 | 7.37 | 1909141621 | 67.44 | 0.04 | 0.19 |
| CW03 | 45.41 | 7.51 | 1901061727 | 66.77 | 1.73 | 0.20 |
| CW03 | 45.41 | 7.51 | 1904230537 | 61.76 | 0.59 | 0.08 |
| CW03 | 45.41 | 7.51 | 1907071508 | 68.42 | 0.86 | 0.10 |
| CW03 | 45.41 | 7.51 | 1910290104 | 65.16 | -0.58 | 0.20 |
| CW04 | 45.37 | 7.61 | 1810292326 | 283.91 | -1.01 | 0.31 |
| CW04 | 45.37 | 7.61 | 1903060013 | 62.58 | 0.44 | 0.09 |
| CW04 | 45.37 | 7.61 | 1907071508 | 68.52 | 1.08 | 0.09 |
| CW05 | 45.32 | 7.73 | 1901260812 | 66.63 | 0.87 | 0.11 |
| CW05 | 45.32 | 7.73 | 1910290104 | 65.35 | -0.71 | 0.09 |
| CW05 | 45.32 | 7.73 | 1911052052 | 189.28 | 1.43 | 0.38 |

Table S2 - Splitting parameters for stations with good SI measurements from at least four different back-azimuthal bin directions. Header: Station | Station latitude (Lat) | Station longitude (Lon) | FPD | FPDerr | TD | TDerr | RMS | Number of Measurements (#)

| Station | Lat | Lon | FPD | FPDerr | TD | TDerr | RMS | # |
|---------|-----|-----|-----|--------|-----|-------|-----|---|
| CE03 | 45.79 | 7.60 | -0.78 | 2.35 | 0.75 | 0.06 | 12.64 | 18 |
| CI07 | 46.23 | 5.52 | 11.58 | 6.57 | 0.13 | 0.04 | 8.05 | 17 |
| CI11 | 46.09 | 6.03 | -0.33 | 1.56 | 0.83 | 0.05 | 13.30 | 9 |
| CI15 | 46.01 | 6.54 | -7.45 | 0.57 | 1.08 | 0.02 | 18.86 | 13 |
| CI16 | 45.92 | 6.62 | -2.58 | 0.66 | 1.09 | 0.02 | 27.78 | 23 |
| CI17 | 45.95 | 6.72 | 11.48 | 0.96 | 0.79 | 0.02 | 17.21 | 17 |
| CI18 | 45.90 | 6.77 | 15.11 | 2.53 | 0.64 | 0.06 | 8.29 | 11 |
| CI19 | 45.94 | 6.90 | 0.98 | 0.61 | 1.32 | 0.03 | 20.85 | 12 |
| CI21 | 45.72 | 7.14 | 37.08 | 2.27 | 1.04 | 0.07 | 18.74 | 15 |
| CI23 | 45.68 | 7.24 | 22.22 | 1.97 | 0.74 | 0.05 | 20.34 | 17 |
| CI24 | 45.65 | 7.25 | 37.75 | 1.59 | 0.93 | 0.05 | 14.58 | 12 |
| CI37 | 44.95 | 8.25 | 87.15 | 12.30 | 0.32 | 0.09 | 9.52 | 7 |
| CI38 | 44.76 | 8.41 | -59.03 | 3.69 | 0.13 | 0.02 | 10.61 | 28 |
| CI39 | 44.67 | 8.47 | -43.54 | 2.80 | 1.01 | 0.07 | 16.59 | 6 |
| CI41 | 44.53 | 8.53 | -53.46 | 0.62 | 0.99 | 0.02 | 19.35 | 7 |
| CI42 | 44.46 | 8.58 | 76.64 | 1.76 | 2.18 | 0.13 | 5.75 | 6 |
| CI43 | 44.40 | 8.62 | -67.28 | 1.86 | 0.85 | 0.06 | 12.82 | 9 |
| CW01 | 45.43 | 7.26 | 1.35 | 3.98 | 0.88 | 0.13 | 12.19 | 8 |
| CW02 | 45.43 | 7.37 | 22.80 | 2.92 | 0.78 | 0.08 | 19.64 | 8 |