# Peer review of "Highlights on mantle deformation beneath the Western Alps with seismic anisotropy using CIFALPS2 data"

_EGUsphere, 2024_

## Author Comment (AC1)

**RC1**: 'Comment on egusphere-2024-468', Anonymous Referee #1, 24 Mar 2024 reply

**Recommendation: Major**

**General comments**

Overall, this is a well-written manuscript which provides an analysis of upper mantle anisotropy along the CIFALPS2 temporary array. Investigating upper mantle anisotropy through shear wave splitting is crucial for comparison with other seismological observables reported for the CIFALPS2 array. The strength of this work lies in its utilization of both conventional shear wave splitting technique (such as tangential energy minimization) and the splitting intensity method. This study adds one more piece of knowledge to mantle deformation beneath the Western Alps. This is quite good organized work with good figures.

Here are my comments to enhance the manuscript and address some unclear points:

**Line 71**

Which criteria were considered by selecting the events? It could be good to mention the criteria here.

*The selection criteria are those described with the sentence "magnitude M>6.0, that occurred between June 2018 and December 2019 and located at a distance interval from the network between 88° and 120°". The amount of events per station, 80 to 150, is due to the different operating time of stations. Some of them recorded for different time windows and consequently recorded a number of earthquakes. Therefore the number of events used are not only related to selection criteria, that were the same all over the entire dataset. We rephrase to avoid any misunderstanding.*

**Lines 80-83**

In addition to plotting all individual splitting measurements at the piercing point of incident SKS ray at 150 km depth, it would be beneficial to plot the back-azimuthal variations of the collected FPD and DT parameters. This could help in interpreting potential complex anisotropy beneath the station and surrounding. These plots can be generated separately for each station or for groups of stations within a 100 km radius. Any notable patterns in back-azimuthal variations should be included in the supplementary material or directly in the manuscript if they provide significant insights.

*Thanks for the suggestion. The amount of measurements per station is not enough, so we plotted the back-azimuthal variations of fast velocity (red dots), nulls (black circles) and time delay for the groups of stations belonging to the 4 sectors identified in the discussion (Figure 5). We include plots only here because the result is not exciting, we could not find any pattern improving our interpretation, mainly due to the poor amount of events coming from the southern quarters back azimuths.*

[Figure]

Furthermore, the back azimuths of null measurements should be marked in the figure of back-azimuthal variations of FPD.

*In the following, here you are the new map with all null measurements marked with different colors with respect to back azimuth. This map will substitute Figure S1 in the Supplementary Material.*

[Figure]

**Lines 101-103**

The precise number of good and fair measurements should be provided quantitatively for clarity. While Figure 2 may offer a visual representation, but providing the exact number will be helpful for the reader.

*Thanks for the suggestion. The amount of good (170) and fair (241) CIFALPS2 measurements has been added in the text.*

**Lines 106-110**

There are too many null measurements more than good or fair quality splitting measurements. That is why, it is necessary to plot all back azimuths of the null measurements of the back-azimuthal distribution of FPD parameters to detailed analyses, as I noted above in the comment for lines 80-83. In a simple anisotropy case, it is expected that the null measurements come only from the slow and fast axis, however, the distributed variations of null back-azimuths may indicate a more complex structure beneath the region. So, it would be beneficial to plot null back azimuths on the graph of back azimuth versus FPD.

*We are aware that we are in a very complex region, also at a small scale, so we are not so surprised by this number of nulls. The same amount of nulls is present in most of the region of Western Alps also in previous papers (i.e. Salimbeni et al., 2018, Tectonophysics; Petrescu et al., 2020, Solid Earth; Link and Rumpker, 2023, JGR), obtained also from permanent stations (light blue data in the background of Figure S1). In Figure S1, the pattern of these measurements is in agreement with the fast directions: for instance, black crosses, coming from 0 and 45 degrees, are present only in the northwestern part of the transect, where a North fast direction is present and somewhere prevailing (sector a and b, Figure 5). The same agreement can be described for the rest of the dataset. We will add some comments also in manuscripts.*

**for TableS2** in Supplementary Material,

**6th Column (FPDerr)**

- I am confused by the presence of negative signs in certain FPDerr values (e.g., for stations CE03A, CI15A). The error for calculated FPD should be equal for both negative and positive part. Furthermore, the significantly high deviations in the average FPD (for example, approximately 80 degrees error for station CI42A) suggest that the number of available SI measurements may be insufficient.

**7th Column (TDerr)**

- In other columns, dots are used to separate decimals, but in the TDerr column, commas are used instead. Is there any special meaning to using commas in this column? Addition to this, given error for time delays are notably high (for example, 1.3sec and 1.7sec TD error reported for the station CI16A and CI42A, respectively).

*Thank you for noticing these inconsistencies. The header was wrong in the previous submission and the last four columns were actually splitting parameters related to possible dipping anisotropy related to a 360 degree fitted sinusoidal curve (see Confal et al., 2023) (and not the errors). This topic is not covered in this manuscript, since the method is not yet sufficiently established and the temporary stations do not have enough measurements for a good fit. Since this topic is not covered in this work, we have decided to delete those rows for simplicity. Instead we have added FPD and TD errors and RMS.*

*The complete and correct header is now: Station Slatitude Slongitude FPD FPDerr TD TDerr RMS NumberMeasurements*

**for Splitting Intensity measurements**,

Considering that the deviations of station-averaged FPD and DT obtained by fitting a sine curve on back-azimuthal variations of individual SI measurements are looking quite large, the reliability of these measurements should be provided to the reader. So, how many individual SI

measurements were used to derive the splitting parameters (FPD and DT) of the station? It should be indicated in the additional column (maybe the 8th column) for the reader to follow.

*An additional column for the amount of SI measurements from which splitting parameters have been calculated has been added as the last column (9th). See prior comment and the header description in the supplementary material for reorganization of columns.*

Furthermore, the root mean square (RMS) calculated from the difference between the fitted sinusoidal curve and each individual SI measurement will be an important parameter for assessing the reliability of the obtained splitting parameters (FPD and DT). This should also be included in the TableS2.

*We have included FPD and TD errors, as well as RMS values for splitting parameter calculations. See supplementary and prior comment.*

**FIGURES**

**Figure 1**

The Ligurian Mountains and Bresse Graben should be highlighted in the first figure, particularly for readers who are not familiar with the study region, as these locations are mentioned in the text but not in the figure.

*We added them. Here you are the new Figure 1:*

[Figure]

*Figure 1 - a) Map of the study region, focusing on the Western Alps. In red are indicated the CIFALPS2 stations, while in blue are permanent and previous temporary stations (i.e. CIFALPS and AlpArray). FPF = Frontal Pennine Fault, BG = Bresse Graben, PP = Po Plain, LM = Ligurian Mountains; b) the red square is the study area reported in a); c) map of all seismic events used in this study, with the star centered in the study region.*

**Figure 2**

Using smaller "**a**" and "**b**" labels to refer to the subfigures can enhance the aesthetics of the figure.

*Thanks for the suggestion. Here you are the new Figure 2:*

[Figure]

**Figure 3**

Here, if certain regions mentioned in the discussion (such as FPF, Ligurian Alps, Po Plain) are indicated above the topographic cross-section in the figure, the reader can follow the results and discussion much more easily.

*Thanks for the suggestion. Here you are the new version of Figure 3:*

[Figure]

**Other Questions**

- What is the amount of the difference between the average splitting parameters (FPDs and TDs) obtained from SWS and SI measurements? If there are any systematic differences between the average anisotropy parameters obtained from SWS and SI techniques, this might be worth adding to the discussion. (The following articles may be helpful, Kong et al., 2015; Monteiller and Chevrot, 2010)

  *The difference between average splitting parameters and values obtained from SI measurements is between 10 and 30 degrees, without any particular pattern. It is true that indications as those found by Kong et al. (2015) need a big amount of data as only permanent stations can give. So, this is a result that in this temporary experiment unfortunately is very rare to be reached.*

- What is the averaging method (misfit surface stacking or basic circular arithmetic mean) by calculating the station-average FPD from individual FPDs obtained from the

tangential energy minimization technique? This should be mentioned in the Data and Methods section.

*To calculate averaged splitting values we used a basic circular arithmetic mean for the fast axes directions and a classic arithmetic mean for the delay time calculation.*

References:

Confal, J. M., Baccheschi, P., Pondrelli, S., Karakostas, F., VanderBeek, B. P., Huang, Z., & Faccenda, M. (2023). Reproducing complex anisotropy patterns at subduction zones from splitting intensity analysis and anisotropy tomography. *Geophysical Journal International*, *235*(2), 1725-1735.

Kong, F., Gao, S. S., & Liu, K. H. (2015). A systematic comparison of the transverse energy minimization and splitting intensity techniques for measuring shear‑wave splitting parameters. *Bulletin of the Seismological Society of America*, *105*(1), 230-239.

Link, F., & Rümpker, G. (2023). Shear‑Wave Splitting Reveals Layered‑Anisotropy Beneath the European Alps in Response to Mediterranean Subduction. Journal of Geophysical Research: Solid Earth, 128(9), e2023JB027192.

Petrescu, L., Pondrelli, S., Salimbeni, S., Faccenda, M., and the AlpArray Working Group: Mantle flow below the central and greater Alpine region: insights from SKS anisotropy analysis at AlpArray and permanent stations, Solid Earth, 11, 1275–1290, https://doi.org/10.5194/se-11-1275-2020, 2020.

Salimbeni, S., M.G. Malusà, L. Zhao, S. Guillot, S. Pondrelli, L. Margheriti, A. Paul, S. Solarino, C. Aubert, T. Dumont, S. Schwartz, Q. Wang, X. Xu, T. Zheng, R. Zhu (2018). Active and fossil mantle flows in the western Alpine region unravelled by seismic anisotropy analysis and high-resolution P wave tomography, Tectonophysics 731, 35-47.

---

## Author Comment (AC2)

**RC2**: ['Comment on egusphere-2024-468'](), Anonymous Referee #2, 31 Mar 2024 reply

In this paper, the authors use seismic data from the CIFALPS2 network to measure shear-wave splitting, thus shedding light on mantle dynamics in the Western Alps. This paper as a whole is well written, shows interesting results and provides reasonable explanations, and the figures are intuitive, thus meeting the interest of *SE*. However, I still have some major considerations for the analysis of shear-wave splitting measurements.

1. the authors assume that the anisotropic source of the measurements is at 150 km and present all the measurements at the pierce points at this depth. Given that subsequent interpretations are based on this, I might suggest that the authors first perform an estimation of the depth of the anisotropic source, e.g., based on the general use of the method proposed by Liu & Gao (2011). https://doi.org/10.1785/0120100258

*Since the scarce depth resolution using the SKS phases, and considering that most of the anisotropy is thought to be in the upper mantle, the depth of 150 km is the one that best approximates its location. To have better details about the depth we followed the suggestion of the reviewer trying to apply the mentioned code. We used it for different cases, testing several dimensions of the region and/or the selection of the measurements, but we didn't find a satisfactory result. As an example, in the following, we show the results of the inversion considering measurements included in what we defined zone A (Figure 5 of the manuscript). No clear visible "valley" is present in the plot and there are no unambiguous depths that could be identified considering different spacing grids. The reason for that unsatisfactory result could be the great variability of fast directions in such a narrow area, that limited the applicability of the code. Liu & Gao, in their paper, apply the code in an area with a very homogeneous fast axes distribution, a condition that is not present here.*

[Figure]

2. Similarly, the authors assumed single-layered anisotropy. Although the actual measured FPDs more or less deviate from surface faults or block boundaries, it is still uncertain whether crustal/lithospheric anisotropy contributes to this, so I would suggest that the authors make a comparison with related studies of crustal anisotropy. SI measurements should be another way in which this can be differentiated, according to Silver & Long (2011). From Fig. S2, monolayer anisotropy is the preferred interpretation and the authors should emphasize it further. https://doi.org/10.1111/j.1365-246X.2010.04927.x

*We did not exclude the presence of a variation of the anisotropy with depth. We consider a valid hypothesis the single-layered anisotropy because of the agreement between mean SWS and SI FPD directions. As shown in Figure 4, most of the averaged directions calculated with the two techniques are in agreement along the chain, some deviations are for stations in the Po Plain, where we know the peculiar difficulty of having high-quality data. As for comparisons with shallower anisotropy measures, we find a good agreement with Pn anisotropic directions (representative of the Moho depth) by Diaz et al. (2013). All this information can confirm our assumption. Moreover, taking into account that the scarce back-azimuthal coverage of our measurements per station hamper the evaluation of the presence of a multilayer structure, even using codes for modelling anisotropic structures like Menke and Levin (2003) or Raysum (Frederiksen, A. W., and M. G. Bostock, 2000), we consider the average values as a main, prevailing signal, that should be related to a principal mantle source. We would better emphasise the concept in the manuscript.*

Some other comments:

Lines 17-19: Please rephrase this sentence.

*Yes, done.*

Fig. 1: Please mark the key block names here. Also please add the scale of the latitude axis, this will help the reader to determine the position (same for the other diagrams).

*Done. Here you are the new Figure 1:*

[Figure]

Lines 45-57: The authors provide an overview of previous studies of lithospheric structure in the region, but for readers unfamiliar with the region, an introduction to the tectonic settings may be missing. In addition, pending scientific issues need further elaboration.

*Thanks for the suggestion. We added a small description of the geodynamic history of the Alps to introduce the following overview of previous studies of lithospheric structure, that helps the understanding of still pending scientific issues.*

Data and Methods: I suggest that the authors show shear-wave splitting measurements with different data quality under different regions in the supplementary material.

*As a working rule, we applied the same criterion of quality assignment to all analysed measurements. This means that a 'good' measurement in Alpine chain and a 'good' measurement in the Po-Plain should have the same properties, that are the same listed at lines 103-105: "The quality assignment is given following the SplitRacer criteria (Reiss and Rümpker, 2017), considering the visibility of the phase, the ellipticity of the initial particle motion and its linearity in the final stage, and the errors associated with phi and dt values."*

*We will include some examples in the Supplementary Material. In the following an example of a "good" measurement:*

[Figure]

*The same consideration is used also for 'null-measurements', i.e. the case in which the energy on the transverse component is absent (see the image in the following) or when a phase did not split because the back azimuth direction of the event is parallel or perpendicular to the anisotropy fast axis direction.*

[Figure]

Fig. 3: R-values are difficult to discern from the graphs. Or maybe use a transparent background colour instead?

*To improve the readability of R values we changed colors (and added also some labels, requested by REV1). Here you are the new Figure 3:*

[Figure]

Fig. 4: Lack of explanation for the dots.

*Dots are the stations. Now we added the explanation in the caption.*

Lines 191-193: Please rephrase this sentence.

*Done*

References:

Díaz, J., Gil, A., & Gallart, J. (2013). Uppermost mantle seismic velocity and anisotropy in the Euro-Mediterranean region from Pn and Sn tomography. *Geophysical Journal International*, *192*(1), 310-325.

Frederiksen, A. W., & Bostock, M. G. (2000). Modelling teleseismic waves in dipping anisotropic structures. Geophysical Journal International, 141(2), 401-412.

Menke, W., & Levin, V. (2003). The cross-convolution method for interpreting SKS splitting observations, with application to one and two-layer anisotropic earth models. *Geophysical Journal International*, *154*(2), 379-392.

Reiss, M. C., & Rümpker, G. (2017). SplitRacer: MATLAB code and GUI for semiautomated analysis and interpretation of teleseismic shear‑wave splitting. *Seismological Research Letters*, *88*(2A), 392-409.